

**The diurnal cycle of cloud profiles over land and ocean between 51°S and 51°N, seen by**
**the CATS spaceborne lidar from the International Space Station**
Vincent Noel [1], Hélène Chepfer [2], Marjolaine Chiriaco [3], John Yorks [4]
1 - Laboratoire d'Aérologie, CNRS/UPS, Observatoire Midi-Pyrénées, 14 avenue Edouard
Belin, Toulouse, France
2 - LMD/IPSL, Sorbonne Université, École polytechnique, École Normale Supérieure, PSL
Research University, CNRS, F-91120 Palaiseau, France
3 - LATMOS/IPSL, Univ. Versailles Saint-Quentin en Yvelines, France
4 - NASA GSFC, Greenbelt, Maryland, USA
Proposed for publication in:
Atmospheric Chemistry and Physics
23 Mar 2018



**Abstract.**

We take advantage of 15 months of measurements from the Cloud and Aerosol Transport System (CATS) lidar on the non-sun-synchronous International Space Station (ISS) to document, for the first time, the diurnal cycle of detailed vertical profiles of Cloud Fraction between 51°S and 51°N. After processing CATS lidar data, we analyzed the diurnal cycles of the cloud profiles over ocean and over continent in two different seasons.

Over the Tropical ocean in summer, the high clouds geometric thickness increases significantly from 1km near 5PM to 5km near 10PM, resulting in a high clouds maximum at nighttime. Over the summer tropical continents, CATS observations reveal the presence of a mid-level cloud layer (4-8 km ASL) persisting all-day long, with a weak diurnal cycle (minimum at noon). Over the Southern Ocean, diurnal cycles appear for the omnipresent low-level clouds (minimum between noon and 3PM) and for the high-altitude clouds (minimum between 8AM and 2PM). Both cycles are time-shifted, with high-altitude clouds following the changes in low-altitude clouds by several hours. Over all continents at all latitudes during summer, the low-level clouds develop vertically and reach a maximum occurrence at about 2.5 km ASL in the early afternoon (around 2 pm).

Our work also show that 1) the diurnal cycles of vertical profiles derived from CATS are consistent with those from ground-based active sensors at local scale, 2) the cloud profiles derived from CATS measurements at local times of 0130AM and 0130PM are consistent with those observed from CALIPSO at similar times, 3) the diurnal cycles of low and high cloud amounts derived from CATS are in general in phase with those derived from geostationary imagery but less pronounced. Finally, the diurnal variability of cloud profiles revealed by CATS strongly suggests that CALIPSO measurements at 0130AM and PM document the daily extremes of the cloud fraction profiles over ocean and are more representative of daily averages over land, except at altitudes above 10km where they capture part of the diurnal variability. These findings are equally applicable to other instruments with local overpass times similar to CALIPSO's, like all the other A-Train instruments and the future Earth-CARE mission.





48     **Outline**

72



## 1. Introduction

Cloud cover diurnal cycles have been documented from space by geostationary satellites as early as the late 1970's (e.g. Gray and Jacobson, 1977) and were summarized based on retrievals from the International Satellite Cloud Climatology Project (ISCCP; Cairns, 1994; Rossow and Schiffer, 1999). Soden (2000) and Tian et al. (2004) used those retrievals to confront the diurnal cycles of clouds, convective activity and water vapor in the upper troposphere, pointing to a clear land-sea contrast. More recently, Philippon et al. (2016) used MSG-SEVIRI data to describe the diurnal variations in the composition of cloud cover above Central Africa. Taylor et al. (2017) also used MSG-SEVIRI to describe when during the day the cloud top temperature is the coldest on average seasonally, over a half-hemisphere grid. Apart from geostationary imagery, few spaceborne instruments provide a sampling frequency well-suited to describe the diurnal variability of clouds. For instance, Wylie (2008) had to take advantage of the four observations per day provided by the NOAA series of polar orbiters to document a weakly-resolved clouds diurnal cycle from multispectral infrared data.

Those studies found the most significant diurnal changes of clouds over continents in summer: low-level boundary layer clouds expand throughout the day, following the warming of the surface by incoming solar radiation, a process significantly affected by orography. In the Tropics, this near-surface activity is transmitted to higher altitudes through deep convection, driving a diurnal cycle in high-level clouds. The time needed for this process to occur delays the cycle of high clouds, whose maximas and minimas occur hours late compared to low-level clouds. At midlatitudes, without deep convection most of the troposphere is free from surface influence, and diurnal changes in the distribution of high-altitude clouds are rather driven by the local atmospheric circulation (e.g. Storm-tracks), leading to less predictable patterns. Over oceans, the largest low-level cloud covers happen in the morning, when the expansion generated by nighttime radiative cooling at cloud top stops. These patterns are supported by understood physical principles and are well documented by passive satellite imagery. But these observations do not provide information on the diurnal cycle of the detailed cloud profiles, which is key to better understand the atmospheric heating rate profile.



Active remote sensing instruments, such as radars and lidars, document the cloud vertical distribution with greater accuracy resolutions than passive instruments, with vertical resolutions finer than 500m. For decades, active instruments have been operated from ground-based sites, building extensive datasets from which time series and statistics about clouds can be derived (e.g. Noel et al., 2006). From space, Liu and Zipser (2008) were able to derive information on the clouds diurnal cycle from the spaceborne Tropical Rainfall Measuring Mission radar, launched in 1997 (Kummerow et al., 1998), but the instrument was not designed to detect clouds with accuracy. The CALIPSO lidar (Cloud-Aerosol Lidar and Infrared Pathfinder Satellite Observations), since its launch into orbit in 2006 (Winker et al., 2010), has provided transformative vertically-resolved data on clouds (Stephens et al., 2017; Winker et al., 2017). Enhanced cloud detections from CALIPSO have, among other things, helped pinpoint and improve significant cloud-related weaknesses in climate models (e.g. Cesana and Chepfer, 2013; Konsta et al., 2016), helped improve estimates of the surface radiation budget (Kato et al., 2011) and of the heating rate profile (L'Ecuyer et al., 2008; Haynes et al., 2013; Bouniol et al., 2016). However, due to its sun-synchronous polar orbit CALIPSO samples the atmosphere at either 1:30AM or 1:30PM local time (LT). The CloudSat radar (Stephens and Kummerow, 2007) and all A-Train instruments (L'Ecuyer and Jiang, 2010) share the same overpass times. Even though measurements limited to two times of day can still offer insights into the day-night cloud changes (Sèze et al., 2015; Gupta et al., 2018), they are insufficient to fully document the diurnal evolution of cloud profiles. This observational blind spot explains why very little is known so far about how the vertical distribution of clouds changes diurnally in most of the globe.

Here we take advantage of measurements from the Cloud Aerosol Transport System (CATS, McGill et al., 2015) lidar on the International Space Station (ISS), to document the diurnal evolution of the vertical distribution of clouds in regions of the globe. The CATS dataset is unique so far, as it contains active vertically-resolved measurements made by lidar from space with variable local times of overpass: the CATS lidar can document cloud profiles at different times along the day between 51°S and 51°N following the ISS orbit.

We first describe how data were selected and processed to derive diurnal cycles of cloud Cloud Fraction profiles and Cloud Amounts from CATS and all other instruments included for comparison (Sect. 2). Then, using CATS retrievals we document, for the first time, the





diurnal cycle of detailed Cloud Fraction profiles in large regions of the globe in two seasons
over ocean and land (Sect. 3.1 and 3.2). In Sect. 3.3 we describe CATS-derived diurnal cycles
of cloud profiles over selected sites and continents with two goals in mind: (i) to compare
them with independent ground-based observations to check the validity of the CATS
retrievals, and (ii) to document the diversity of the continental cloud profile diurnal cycles
over the globe. In Section 4 we discuss implications of our results: We compare the diurnal
cycle of the Low and High cloud covers derived from CATS with ones from geostationary
satellites (Sect. 4.1), and discuss the agreement between CATS Cloud Fraction profiles
derived at the times of CALIPSO overpass with actual CALIPSO retrievals (Sect. 4.2.a). Finally,
we consider CATS profiles at overpass times from current and future sun-synchronous
spaceborne lidar missions (Sect. 4.2.b) to understand which part of the diurnal cloud cycle is
sampled by these instruments. We conclude in Sect. 5.



## 2. Data and Methods

### 2.1 Data

#### *a) Cloud detections from the CATS spaceborne lidar*

In this study our primary data consist of cloud detected during June-July-August (JJA) and December-January-February (DJF) periods using data from the CATS lidar system (Yorks et al., in preparation). CATS operated from the ISS between February 2015 to late October 2017. Although CATS was originally designed to operate at 3 wavelengths (355, 532 and 1064nm) with variable viewing geometries, beginning in March 2015 technical issues limited operation to a single 1064nm wavelength and a single viewing mode. The CATS instrument went on providing single-channel high-quality data (Yorks et al., 2016a) until a fault in the on-board power and data system ended science operations on October 30, 2017.

Being located on the ISS means measurements from CATS are constrained to latitudes below 51°, giving it access to ~78% of the Earth's surface. This however prevents our study from covering polar regions. However, this leads to densely distributed overpasses at latitudes above 40°: CATS sampling is particularly good in populated midlatitude regions and above the Southern Ocean.

CATS cloud detections were derived from vertical profiles of ATtenuated Backscatter measured every 350m at 1064nm with a 60m vertical resolution. ATB profiles were calibrated, processed and averaged based on the procedures designed for CALIOP data to enable threshold-based cloud detection (Yorks et al., 2016b and in preparation). Unlike for CALIOP, the cloud detection algorithms for CATS rely primarily on 1064nm data. They create the CATS operational Level 2 (L2) products, which provide properties for detected clouds (including base and top) every 5km along-track. Hereafter we used such cloud properties from CATS L2O data files v2.01 (Palm et al., 2016), including only layers with a Feature_Type_Score above 5, to avoid including wrongly-classified optically thick aerosol layers near deserts. To document the diurnal cycle (Sect. 2.2.a), we used data obtained in both nighttime and daytime (sunlit) conditions between March 2015 and October 2017.



CATS cloud data being still novel at the time of this writing, we document and discuss
several of its characteristics in Appendices A and B, including sampling variability and the
sensitivity of cloud detection in presence of solar pollution. This exploration of CATS data
(and the upcoming comparisons with other instruments) made us confident that its
sampling and cloud detections are robust enough to be used for scientific purposes.

*b) Cloud detections from ground-based active instruments*

Like with any lidar, the CATS laser beam gets fully attenuated when passing through clouds
with optical depths larger than typically 3 (e.g., Chepfer et al., 2010). This can lead to the
cloud fraction being underestimated in the lower troposphere. To estimate how much the
CATS cloud fraction is biased at low altitudes, we compare CATS detections with
independent observations collected from ground-based active instruments.
Ground-based observation sites provide long-term records of atmospheric properties over
periods that often cannot be reached by satellite instruments (Chiriaco et al., in revision).
Nowadays such sites are often well equipped with active remote sensing instruments. Data
acquisition, calibration and processing are often homogenized in the framework of specific
observation networks (e.g. EARLINET, the European Aerosol Research Lidar Network,
Pappalardo et al., 2014). Descriptions of the clouds diurnal cycle based on ground-based
measurements are however scarce. In this study, we compare CATS cloud cycles with those
derived from active measurements at two ground-based sites in Europe and the United
States.
The first ground-based site is the Site Instrumenté de Recherche par Télédétection
Atmosphérique (SIRTA, Haeffelin et al., 2005), 20km South-West of Paris at 48.7°N, 2.2°E.
From SIRTA we used observations from the Lidar Nuages et Aérosols (LNA, Elouragini and
Flamant, 1996). Cloud detections from the LNA were homogenized, repackaged and made
available in the framework of the SIRTA-reOBS project (Chiriaco et al., 2014; Chiriaco et al.,
in revision). The LNA requires human supervision and does not operate under precipitation,
leading to irregular sampling and almost no nighttime measurements. Its long operation
time however means its dataset covers almost 15 years. Cloud layers were detected in LNA
profiles of attenuated backscatter following a threshold-based approach similar to CATS and





CALIPSO.
The second site we consider is the Atmospheric Radiation Measurement (ARM) Southern
Great Plains (SGP) site, at 97°W, 36°N. From this ground-based site we consider cloud
retrievals from the Millimeter Wavelength Cloud Radar (MMCR) and from the Raman Lidar
(RL). The MMCR has been routinely operated to detect and identify clouds and
precipitations since 1996 (Moran et al., 1998), while RL cloud detections are available since
1998 (Ackerman and Stokes, 2003). In the framework of the present study we did no specific
processing of data from these instruments. Instead, we compare CATS cloud retrievals over
the SGP site with the descriptions made by Zhao et al. (2017, Fig. 3a) and Dupont et al.
(2011, Fig. 3) of the diurnal cycle of clouds over SGP based on 14 years of MMCR cloud
detections and 10 years of RL cloud detections in Sect. 3.3.

***c)   Cloud detections from passive and active spaceborne sensors***
In addition to the CATS, LNA and MMCR datasets, in the upcoming sections we use cloud
retrievals from two spaceborne datasets to put CATS cloud retrievals into a referenced
context. First, we consider the baseline reference for the description of the clouds diurnal
cycle from space: the analysis of data from the ISCCP done by Rossow and Schiffer (1999),
hereafter RS99. Their results are based on aggregated and homogenized infrared and visible
radiances from imaging radiometers on the international constellation of weather satellites.
They are widely considered as the reference for describing the diurnal cycle of the cloud
cover at large scales from space measurements. Like with SGP data, we did not reprocess
any ISCCP data for the present study, instead we rely on the description of the diurnal cycle
of low and high clouds RS99 documented in their Fig. 11 based on ISCCP, to which we
confront CATS retrievals in Sect. 4.1.
Finally, we also confront CATS cloud detections with retrievals based on measurements
from the CALIOP lidar, routinely made since 2006 from the sun-synchronous CALIPSO
platform at 13:30 and 01:30 LT in Sect. 4.2. To enable comparison with CATS retrievals, we
used cloud layers retrieved from CALIPSO measurements during the period of CATS
operation (March 2015 to October 2017), and documented at 5km horizontal resolution in
CALIPSO Level 2 V4.10 Cloud Layer Products (Vaughan et al., 2009). We processed both



CATS and CALIPSO data alike as described in Sect. 2.2.a.


**2.2. Methods**

*a)  Building the diurnal cycle of Cloud Fraction profiles from lidar cloud detections*

Analyzing CATS lidar echoes lets one identify at which altitude a cloud is present above a
particular location on Earth at a given moment. By aggregating such information over a long
period, vertical profiles of Cloud Fraction (CF) can be derived. A CF(z) profile documents at
which frequency clouds were observed at the altitude *z* over a particular location. Cloud
Fractions are conceptually equivalent to the Cloud Amounts derived from passive
measurements (next section), but vertically resolved with a 60 meters resolution.
From CATS level 2 data files, we extract profile-based cloud detections and use the
measurement UTC time and coordinates to deduce their local time of observation. Using the
resulting list of cloud layer altitudes, coordinates and local times of detection, we count the
number $n$ of cloud detected within half-hour bins of local time, 2°x2° lat-lon boxes and
200m altitude bins. We also count the number of valid data points $n_0$ within those bins.
Eventually we derive the Cloud Fraction $CF = \frac{n}{n_0}$, either in individual local time/lat-
lon/altitude bin or by aggregating $n$ and $n_0$ over a selection of bins.
CATS reports cloud layers as opaque when no echo from the surface is found in the profile
below a detected cloud, following the same methodology as in Guzman et al., 2017. Below
an opaque cloud layer, there is no laser signal left to propagate, and clouds potentially
present at lower altitudes will not be sampled by the lidar. To account for this effect, we
consider the portions of profiles below an opaque layer unsampled, and they do not count
in the number of valid data points $n_0$. This approach limits the influence of laser attenuation
on cloud detections but cannot totally cancel it.
To enable comparisons with CATS CF profiles (Sect. 3.3 and 4.2), we followed a similar
approach to build CF profiles using cloud detections from SIRTA-reOBS (Sect. 2.1.b) and



from CALIPSO Level 2 products (Sect. 2.1.c). In both cases, we counted the number of cloud
detections and valid (non-attenuated) measurements in hourly local time bins and 200m
altitude bins. For CALIPSO, only 01:30AM and PM time bins were filled.

***b) Building the diurnal cycle of Low and High Cloud Amounts from CATS data***

As ISCCP data are based on radiances, clouds therein are characterized according to

their retrieved top pressure P as low (P > 680hPa), middle (440 < P < 680hPa) or high
(P < 440hPa). To enable a direct ISCCP-CATS comparison, we derived Cloud Amounts (CA)
from CATS data for low and high clouds as defined by altitude: low clouds have their base
below 4km ASL, high clouds have tops above 7km, and mid-level clouds are in between.
Using the list of cloud layer altitudes, coordinates and local times of detection derived from
CATS detections (Sect. 2.2.a), we count the number of occurrences $n'$ of at least part of one
cloud layer in half-hour bins of local time, 2°x2° lat-lon boxes and the three altitude ranges
(0-4km, 4-7km and higher than 7km ASL). We also count the number of occurrences $n'_0$ that
could possibly be reported given the measurements sampled by CATS within each bin,
taking into account the existence of opaque layers. Eventually, we derive the Cloud Amount
$CA = \frac{n'}{n'_0}$ for low, mid and high-altitude clouds layers, either in individual local time/lat-lon
bin or by aggregating $n'$ and $n'_0$ over a selection of bins. Like RS99, we separated CATS cloud
detections over land and ocean, based on the International Geosphere-Biosphere
Programme surface flag present in CATS L2 products on a profile basis (Palm et al., 2016).



## 3. Results

### 3.1. Diurnal Cloud fraction profiles observed at Global scale

Figure 1 shows the global diurnal cycle revealed by CATS data over Ocean and Land during JJA from March 2015 to October 2017. Low and high clouds are clearly separated, with a band of minimum cloudiness in-between (near 4km Above Sea Level or ASL). Above both surfaces CATS data show large amounts of high clouds during nighttime that get thinner near noon as their base rise. The vertical evolution in the fraction of sampled atmosphere due to attenuation by atmospheric components, for these diurnal cycles and all that follow, is documented in Appendix C.

Significant differences exist between the cloud profiles diurnal cycle above land and ocean. Clouds generally extend higher over land during nighttime: high clouds are vertically most frequent near 10km over ocean, while they extend up to 14km above continents until 5AM. Over ocean, high clouds appear to rise late in the afternoon (3-6PM) and fall soon thereafter as night falls. Land-ocean differences are most striking at low altitudes: over Ocean low clouds are present almost all day long between 0 and 2km ASL, their CF decreasing from a 20% maximum near 4AM to ~10% between 11AM and 5PM. Over land, low clouds are only significant during daytime: they appear near 2km ASL at 10AM and extends upwards to reach 4km ASL near 4PM. The associated CF remains low, at most 8%. These planetary boundary layer (PBL) clouds are most certainly associated with turbulence and convection activity occurring near the surface. They disappear after 4PM without connecting to the higher layers. The clear-sky band (CF < 2%) near the surface is thickest at night (almost 2km) and thinnest in the late morning.

An aside on cloud detection: over the ocean, CATS detects both low and high clouds more frequently during nighttime. This suggests that the high clouds are optically thin enough for letting CATS document the increase of lower clouds. If the reverse was true, more high clouds would be systematically linked to fewer low clouds, which is not what we observe. The frequency of high-level clouds observed in daytime could however be affected by the decrease in cloud detection sensitivity due to solar pollution affecting the signal to noise



ratio. While CATS is seeing the diurnal cycle of high and low clouds, the magnitude of the
daytime cloud fractions could then be biased slightly low due to solar pollution and, at low
altitudes, cloud-aerosol discrimination issues.
While these seasonal mean results are informative, they mix together unrelated cloud
populations from hemispheres with opposite seasons driven by different circulation
regimes. We thus describe the daily cycles of clouds in zonal bands in the next section.



### 3.2. Diurnal Cloud fraction profiles observed over mid-latitudes and Tropics

In this section we consider cloud populations over four latitude bands: midlatitude (30°-51°) and Tropics (0-30°), in the North Hemisphere (NH) and South Hemisphere (SH), over land and ocean. We first examine the differences between the diurnal cycles affecting the cloud vertical profiles over ocean and land in JJA (Sect. 3.2.a and 3.2.b, Fig. 2), then we discuss how these cycles are affected by the season by considering DJF results (Sect. 3.2.c, Fig. 3).

*a) High clouds*

As expected, Fig. 2 shows that high clouds are located at higher altitude in the tropics (12-16km ASL) than in midlatitude (8-12km), following the variation of the troposphere depth with latitude. Also as expected, the occurrence of high clouds is largest (CF > 20%) in deep convection along the Inter Tropical Convergence Zone (ITCZ), located between 0° and 30°N in JJA, and minimum (CF < 8%) in the subsidence branch of the Hadley cell (0°-30°S in JJA). In mid-latitudes, high clouds (7-9km ASL) are far more frequent (CF ~ 20%) over the Southern Ocean (30°S-51°S) than over the northern ocean (30-51°N).

Oceanic high clouds CF exhibits a marked maximum at nighttime and a pronounced minimum at midday in all latitude ranges (tropics and mid latitudes). Even if this strong diurnal cycle occurs at all latitudes (even in subsidence region), it is more pronounced where the high clouds are more numerous: along the ITCZ (0-30°N) and in the Southern Ocean (30-51°S). In addition to the variation in the high cloud occurrence, the vertical extent of these clouds shows a marked diurnal cycle as well along the ITCZ: more than 4km near midnight, less than 1km at noon. This thickening takes a few hours (5-10PM), while the morning thinning is much sharper. By comparison, over the Southern Ocean the thickness of high clouds remains quite stable throughout the day.

Overall, high clouds behave very similar above land (Fig. 2, right column) and ocean (Fig. 2, left column) at all latitudes, except between 30-51°S where land surface is too small to conclude.

*b) Low clouds*



Over ocean in JJA (Fig. 2), the occurrence of low clouds (0-3km ASL) changes significantly
with latitude: The Southern Ocean region (30-51°S) is by far the cloudiest, the mid-latitude
north (30-51°N) and the subsidence tropics (0-30°S) are moderately cloudy, and even less
low clouds are observed along the ITCZ (0-30°N). The oceanic low clouds show only small
variations along the day. A weak diurnal cycle occurs at all latitudes except along the ITCZ
(possibly because there the low clouds are in part masked by higher clouds affected by an
out-of-phase diurnal cycle). Low-level clouds are more numerous in nighttime (CF near 20%)
compared to daytime (CF~12%) in subsidence tropics (0-30°S) and mid-latitude north (30-
51°N). The southern oceanic low clouds exhibit a very faint diurnal cycle: their CF gets over
20% nearly all day long, with a very small decrease near 2PM.
In contrast to high clouds, the differences between land and ocean are striking for the low
and mid-level clouds. Both the occurrences and the diurnal cycles of clouds over land differ
significantly from their oceanic counterparts. The low clouds are very few over land (CF~4%)
compared to over ocean (>16%), all day long. Moreover, the continental low cloud diurnal
cycle exhibits a maximum in the early afternoon (around 2PM) that does not show up over
ocean: a maximum CF appears around 2.5 km of altitude in the upper edge (or just above
the top) of the atmospheric boundary layer; it is linked to convective activity between 10AM
and 5PM.
Another noticeable difference between land and ocean is the presence of well-defined mid-
level cloud population over NH tropical land (0-30°N, 2nd row on the right in Fig. 2) in the
free troposphere between 5 and 7 km ASL. These mid-level clouds show a diurnal cycle
opposite to PBL clouds and similar to the high clouds in that its minimum occurs at midday
and its maximum at night, although the magnitude of this cycle is much more limited.
Bourgeois et al. (2017) discussed similar clouds observed over West Africa: they found these
clouds reach maximum occurrence early in the morning, which is consistent with our results.

*c) Seasonal differences*
Figure 3 presents diurnal cycles of cloud fraction profiles over the same latitude bands as
Fig. 2 but based on data collected during the boreal winter (DJF). As seasons switch
hemispheres, we anticipate cloud populations to undergo symmetric changes across





hemispheres, in agreement with large-scale dynamic processes driving their spatial
distribution on seasonal time scales. This is verified for high clouds (Fig. 2 vs. Fig. 3): in the
Tropics the ITCZ moves to South and with it the large CF at high altitudes, in midlatitudes the
high clouds are more frequent during the winter season, due to more frequent low-pressure
conditions.
Interestingly, the mid-altitude clouds visible near 6km ASL in the NH Tropics over land (Fig. 2,
2nd row on the right) also move to the SH Tropics in DJF (Fig. 3, 3rd row on the right). This
confirms the year-long persistence of midlevel clouds over continental tropical regions found
by Bourgeois et al. (2017).
The seasonal changes in low clouds are less symmetric than in higher clouds, as they are
more closely related to surface conditions. Over ocean, in DJF the amount of low clouds
increases dramatically in NH midlatitudes compared to JJA (Fig. 2 and 3, top left), but does
not change noticeably in the SH midlatitudes: the diurnal cycle that sees a slight decrease in
the huge population of low clouds over the Southern Ocean is present in both seasons (Fig.
2 and 3, bottom left). Over land, in the Tropics, low clouds appear similar in frequency and
behaviour in both DJF and JJA: PBL clouds extend vertically between ~7AM to 5PM (Fig. 2
and 3, rows 2 and 3 of right column). The NH midlatitudes show the strongest seasonal
change in low clouds, as they become present all day long: the diurnal cycle associated with
PBL development in JJA disappears in DJF (Fig. 2 and 3, top right). SH midlatitude retrievals
over land are as noisy in DJF and JJA, but the DJF data (Fig. 3, bottom right) suggests that low
clouds there extend vertically a lot more than in JJA, up to 4km ASL.





### 3.3. Diurnal cycle of cloud profiles above selected continental regions

In this section, our first goal is to compare the diurnal cycle of the cloud fraction profiles from CATS against independent observations collected by active instruments from ground-based sites (Sect. 3.3.a and 3.3.b). In particular, we want to understand if the results shown so far (Fig. 1-3) are valid for low clouds despite the attenuation of the space laser signal (Sect. 2.2.a). Our second goal is to compare, for the first time, the diurnal cycle of the cloud fraction profiles over different continental regions all over the globe as observed with a single instrument (Sect. 3.3.c).

### a) Over South of Paris in Europe

Figure 4 shows the diurnal evolution of CF profiles seen by the ground-based LNA lidar (Fig. 4a) operated on the SIRTA site south of Paris (Sect. 2.1.b) and seen by CATS space lidar in a 10°x10° box centered on the same site, keeping only profiles sampled over land (Fig. 4b). Both datasets report a well-defined high-altitude layer, with a clear-cut cloud top near 12 km ASL that rises up a few hundred meters in the morning until 10AM and slowly falls during the afternoon by at most 1 km. In both figures the bottom of this layer is not sharply defined: the CF decreases almost linearly from 20% near 11-12km ASL to near-zero at 4km ASL. Both instruments also report a low-level cloud layer that extends upwards from ~1km ASL at 5AM to ~4km ASL near 8PM.

Regarding differences, the space lidar sees a late-afternoon resurgence of high-altitude clouds (starting near 5PM) absent from the ground-based lidar record. The space lidar also reports a much lower frequency of boundary layer clouds: less than 10% throughout the day. This difference gets particularly large in the late afternoon, when the ground-based lidar reports the low-level CF rising above 20%. The large quantity of high-altitude clouds observed by CATS at that time could impair its ability to detect lower clouds, while at the same time the large quantity of low clouds observed by the ground lidar can impair its ability to detect high clouds. The absence of precipitating clouds from the LNA dataset could also explain this difference.

As expected, the spaceborne CATS lidar sees more high-level clouds and less low-level



clouds than the ground-based LNA lidar . This sampling bias affects all ground-space lidar
comparisons (e.g. Dupont et al., 2010). Even so, the diurnal cycle of the cloud altitudes are
roughly consistent from space and ground lidars. This comparison suggests the main
limitation of CATS is the capability to document the increase in low cloud occurrence in the
late afternoon.

**b) Over the US Southern Great Plains ARM site**
Figure 4c shows the diurnal evolution of CF profiles based on CATS measurements in a
10°x10° lat-lon box centered on the ARM SGP site (Sect. 2.2.b), keeping only profiles
sampled over land. High-level clouds near 12km Above Sea Level (ASL) are frequent during
nighttime, with large CF (above 20%) between 16:00 and 03:00 LT. The high cloud layers
also get thick and extend vertically between 9 and 14km ASL. The importance of high-level
clouds strongly drops during daytime (7AM-5PM), with CF dropping below 10% at midday.
The associated cloud layer gets much thinner, limiting its extent between 11 and 12km ASL
at its thinnest point (near 10AM). There are slightly more midlevel clouds (4-8km ASL) in the
early morning, with CF increasing to ~10% between midnight and 7AM. Midlevel clouds are
almost non-existent the rest of the day. PBL clouds form near the surface at 9AM, rise and
thicken almost up to 4km ASL near 4PM.
Most of these features derived from CATS observations are consistent with those derived
from summer observations by the ground-based MMCR (Fig. 3a, top left in Zhao et al., 2017)
and RL (Fig. 3, bottom right in Dupont et al., 2011 -- mind the x-axis in UTC, which brings the
local noon at 18UTC). For instance, Both CATS and the ground-based datasets report a
rather strong diurnal cycle of high clouds, which can be explained by possible influence from
Tropical dynamics at the 36°N latitude of the SGP site. There are some differences: both
MMCR and RL report a minimum in the high-level clouds near 5PM. The MMCR reports a
thinner extent of boundary layer clouds (up to 2.5km at most), while findings from the RL
are more consistent with those from CATS. The MMCR reports almost no low-level clouds
between 6PM and 6AM, while CATS and the RL report some clouds in that time frame --
they might be optically thin and missed by the radar. The RL reports almost none of the
daytime PBL clouds so conspicuous in MMCR and CATS observations, perhaps because fully
attenuating clouds were removed from the RL dataset for the Dupont et al. (2011) study as



they hide most of the atmosphere from a ground-based lidar (unlike a spaceborne one).
These deviations appear acceptable to us given the much smaller size of the CATS
dataset (infrequent overpasses over 3 seasons) compared to the daily local measurements
included in the MMCR and RL datasets (14 and 10 seasons). It is reassuring to find that CATS
results retain the major features of the clouds profile daily cycle. Most notably, CATS
provides a correct representation of the diurnal evolution of the altitude of low-level
boundary layer clouds (not the occurrence in late afternoon) despite the presence of high-
level clouds.
In this section we have seen that using retrievals from ground-based instruments as a
reference, CATS measurements seem to provide an interesting documentation of the clouds
diurnal cycle. Due to the distribution of ground-based sites, however, this validation
approach is limited to certain regions: mostly midlatitudes from the Northern Hemisphere.

***c) Diurnal cycles of the cloud profiles over continents***
Continents are diverse in ground type, orography, latitude, exposition to large scale
atmospheric circulation, and transport of air masses from the local environment. These
factors influence the atmosphere above the continent, leading to possible variations in the
cloud diurnal cycle profiles. Ground-based observations let us document these different
cycles, but differences between instruments and operations in the different ground sites
make comparing diurnal cycle observed from ground at different locations difficult. Thanks
to CATS data, for the first time we compare here the cloud diurnal cycle profiles observed
over different continents by a single instrument and with a relatively large space sampling,
compared to single-site ground-based observations. Figure 5 illustrates how the diurnal
cycle of CF varies among seven large continental areas across both hemispheres, considering
only cloud detections made by CATS over land within lat-lon boxes (defined in the inset map)
during the summer seasons (JJA in the NH, DJF in the SH).
During summer most continents share a development of PBL clouds during sunlit hours
(with similar Cloud Fractions, hours and vertical extents), except NH Africa where low clouds
are almost absent. Most continents also share a nighttime maximum and daytime maximum
of high clouds, with an associated thinning during morning and thickening during the



afternoon. Variations in cloudiness and cloud thickness are particularly intense over South
America and SH Africa, while they are minimal over Australia. A mid-altitude cloud layer is
present almost all day long, with a faint daytime minimum, over all SH continents and NH
Africa.
Note that the present comparison is less robust in the lower troposphere than higher in the
troposphere, due to the attenuation of the space lidar signal as it penetrates the
atmosphere.



**4. Discussion**

Hereafter we use our results for answering the following questions: How does the diurnal
cycle of low, mid, high cloud covers from geostationary satellites compare with CATS ones?
Do the existing lidar space missions document extreme or average behaviours of the cloud
profile diurnal cycle? What about upcoming sun-synchronous lidar space missions?

**4.1 About the Diurnal cycles of Low and High Cloud Amounts**

CATS observations provide a first opportunity to compare the cloud diurnal cycle derived
from the ISCCP dataset (Sect. 2.1.c) with completely independent observations at near-
global scale (excluding latitudes higher than 51°). In particular, we expect an active sensor
technique (CATS) to be independent of the surface, contrarily to the passive remote sensing
observations (ISCCP) that may sometimes confound clouds and surface over reflective
surfaces such as ice and deserts. Moreover, CATS is expected to observe more optically thin
clouds than passive sensors thanks to a lidar high sensitivity. Since CATS sampling is
constrained between 51°S and 51°N, its data cannot be used to document the diurnal cycle
in the polar regions, like ISCCP does: our comparison will extend at most to midlatitudes.
Figure 6 shows the diurnal cycle of the Low and High cloud covers observed by the CATS
space lidar, plotted in a similar way as Figure 11 in RS99 for easier comparison.
Over ocean CAs are very stable, the diurnal cycle is almost flat (Fig. 6, left column). CATS
shows a weak cycle for low clouds, with a maximum in mid-morning and a minimum in
early-afternoon, which is also visible in ISCCP data. For oceanic high clouds, CATS exhibit
almost no diurnal cycle except in the Tropics where they follow the same cycle as low
clouds. ISCCP also shows a weak cycle for high clouds, but opposite to the CATS one. This
might be related to the fact that CATS can detect optically thin high clouds better than
ISCCP. The optically thicker high clouds seen by ISCCP are thus probably more linked to deep
convection activity. CATS can better detect optical thin high clouds, which should be more
decoupled from convection and less affected by diurnal cycles.
Over land, between 15°S and 60°N, CATS reports that low-clouds have a pronounced diurnal



cycle with a maximum of low-level clouds at midday (+10%) and a minimum at midnight (-
5%). This is consistent with ISCCP observations (Figure 11 in RS99), but in the Northern mid-
latitudes the amplitude of the cycle is weaker for CATS than ISCCP (minimum at -4% instead
of -12%).  For high-level clouds over land in the Tropics (15°S-30°N) CATS observes a
maximum during night-time and a minimum at noon; the timing is consistent with ISCCP but
the amplitude is slightly more pronounced with CATS than ISCCP (-12% instead of -7% at
midday). In the Southern hemisphere (15°S-60°S) the similarity between CATS and ISCCP
gets lost, probably because the land surface is small in those latitude ranges and the
observations are not significant.
In summary CATS confirms the shape of the Low and High cloud diurnal cycles observed by
ISCCP except for high tropical clouds, likely because the space lidar can detect more
optically thinner clouds that are not directly linked to deep convection. In most cases, the
amplitudes of the diurnal cycle observed by CATS differ from those observed by ISCCP.
Both CATS and ISCCP miss some low clouds that are masked by the presence of high thick
clouds. So even if CATS and ISCCP diurnal cycle are roughly consistent in low clouds, both
results might be biased in the same direction. The high clouds diurnal cycle presented here
are more robust than the low clouds ones.

**4.2 About the Cloud Fraction profiles observed at fixed local times by space lidars**
The CALIOP lidar has provided detailed Cloud Fraction profiles since 2006 at 0130AM and
0130 PM LT. The next spaceborne atmospheric lidar missions ADM-Aeolus, to be launched
in late 2018 (Culoma et al., 2017) on a sun-synchronous orbit, will enable measurements at
0600AM and 0600PM LT. After that, the ATLID lidar on the Earth-CARE platform (Illingworth
et al., 2015), expected to launch in 2020, will operate at fixed local times close to CALIOP
(02:00AM and PM). The CATS dataset may remain for the near future our single source of
diurnally distributed cloud profile lidar measurements from space.

*a) Comparison between CATS and CALIPSO*
In this section we first check how CATS sees the day/night variation in cloud profiles also



documented by CALIOP through its two daily overpasses. Figure 7 shows vertical profiles of
Cloud Fraction reported by both datasets at 0130AM and PM, over ocean (left) and land
(right), latitude-weighted and averaged between 51°S and 51°N over JJA between 2015 and
2017. The black lines show the CF obtained when considering all measurements from both
instruments. Over land and ocean, we find that both CALIPSO and CATS overall report larger
Cloud Fractions at 0130AM (blue) than 0130PM (red), in agreement with the findings of
Gupta et al. (2018). Below 2.5 km, this difference is stronger over ocean (+7% in 0130AM
CF) than over land. Both datasets report a strong increase in 0130AM CF (almost +7%
compared to 0130PM) above 15km over land.
The CF profiles reported by both datasets agree very well over Ocean (left) in both daytime
and nighttime conditions. Over land (right) in daytime (red) conditions, CATS reports slightly
more low-level clouds (CF~7% near 1km ASL, ~5% for CALIOP). This difference, which is
present at all latitudes above land during daytime (not shown), might be due to the so-
called single-shot low clouds, for which CALIOP data undergoes a specific processing
(Winker et al., 2009). The strongest differences appear for nighttime CF over land (right,
blue): CALIPSO CF is larger than CATS CF by a 2-3% throughout the entire profile. A perfect
agreement between CF from both datasets should not be expected, as the CATS and CALIOP
lidars operate in different configurations – wavelengths, pulse repetition frequencies and
signal-to-noise ratios are different, for a start. These technical variations lead to differences
in, for instance, how fast the laser pulse energy of both instruments gets attenuated as it
penetrates atmospheres of various compositions, or differences in cloud detection
performance, e.g. when sampling optically thin clouds in the upper troposphere, or
fractionated boundary layer clouds (see Reverdy et al., 2015 for a study of the impact of
design choices on lidar retrievals). Both datasets agree quite well on the general vertical
pattern of the profile, though. A useful conclusion is that considering CALIPSO observations
at both overpass local times (i.e. 0130AM and 0130PM) apparently provides a good
approximation of the daily average Cloud Fraction profile.

b) Comparison of cloud fraction profiles at various times of satellite overpass
As a final analysis, we represent the range covered by CATS hourly Cloud Fraction profiles
over a day (averaged over the globe - white envelope in Fig. 8) and show CF profiles



observed by CATS ±1 hour around the fixed local observation times of the three sun-
synchronous space lidar missions (CALIPSO, ADM-Aeolus, EarthCare).
Our first aim is to understand how wind observations made at fixed local time by ADM-
Aeolus might be impacted by the cloud diurnal cycle. ADM-Aeolus will provide information
on wind only in absence of clouds. Figure 8 indicates that ADM-Aeolus overpass times are
quite cloudy in both AM and PM compared to the diurnal variability (white envelope). The
PM overpass corresponds to daily maximum in cloud profiles over both ocean and land,
while AM observations correspond to a time representative of the daily average cloud
fraction profile. As more clouds occur in the PM than AM observations, less wind
information will likely be provided by ADM-Aeolus in the afternoon than in the morning. For
the future, another ADM-Aeolus-like mission around midday (minimum cloud fraction
profile) would increase the number of wind measurement with respect to the cloud
occurrence.
Our second aim is to understand how well observations made at fixed local times by space
lidar dedicated to clouds studies (CALIPSO and EarthCare) capture the daily variability of
cloud fraction profiles. Figure 8 suggests that over land (right), CALIPSO and Earth-CARE
retrievals capture only part of the daily CF variability above 8km ASL: the PM measurements
overestimate the daily CF minima and the AM measurements underestimate the daily CF
maxima. Below 8km ASL they are rather representative of the daily average, except below
5km ASL where PM measurements get close to the daily CF maxima. Figure 8 also shows
that over Ocean (left) CALIPSO and Earth-CARE retrievals should be considered as the daily
CF maxima during the nighttime (AM) overpass and as the daily CF minima during the
daytime (PM) overpass. This has interesting implications: it suggests that not only CALIPSO
but all the observations dedicated to cloud studies collected by the instruments within the
A-train (CloudSat, CERES, MODIS, PARASOL, etc.) have documented the state of the
atmosphere in the extreme states of the cloud profile diurnal cycle over the last 12 years
over ocean. These conclusions suggest the A-Train observations are likely relevant and
robust to constrain the cloud diurnal cycle extremes in climate models and climate studies.






### 5. Conclusions

In this paper we took advantage of the variable local time of overpass of the International
Space Station to document the diurnal cycle of the cloud vertical profile as seen by the CATS
lidar. This is the first time the diurnal evolution of the vertical cloud profile is documented on
that vertical scale on a large part of the globe, between 51°S and 51°N. Our results are based
on 15 months of systematic observations (3 boreal summers and 2 austral summers)
collected during the 2015-2017 time period, which enable statistically significant results.

The main results are the following. We observed the high tropical clouds start getting thicker
late in the afternoon (4-5PM) and reach their maximum thickness of 4-5km near 10PM. This
thickening is particularly intense in the Summer Hemisphere in DJF. Our results reveal a mid-
level cloud layer (4-8 km ASL) persistent all day long over the tropical continent during
summer, with a weak diurnal cycle (minimum at noon). Southern Ocean results are quite
unique; this ocean is covered by low clouds (0-2km ASL) all day long in summer and winter. A
slight diurnal cycle sees their CF drop by a few percents during the afternoon (from noon to
6PM), but their thickness stays constant. High clouds are also frequent over the Southern
Ocean, more so in JJA, and follow all year long an earlier diurnal cycle, with an early morning
to early afternoon minimum (from 8AM and 3PM). At all latitudes, continental low clouds
are most frequent in the early afternoon (around 2PM) at about 2.5 km ASL. Our results also
show that the diurnal cycle of clouds in summer share similar features over continents in
both hemispheres: the rapid development of near-surface clouds during sunlit hours and an
increase in cloudiness and cloud thickness at high altitudes during nighttime (stronger over
the SH and the Tropics). Exceptions are NH Africa, where PBL clouds are very rare, and
Australia, where high clouds appear only significant between 8 and 11PM.

We evaluated the diurnal cycle derived from CATS against independent ground-based
observations and found satisfactory agreement. Moreover, we discussed the implications of
our results for spaceborne instruments from sun-synchronous satellite missions (CALIPSO
and the A-train, ADM, Earth-CARE). Our results suggest that cloud profiles from CALIPSO and
Earth-CARE over oceans should nearly describe the daily minimum of the cloud fraction



vertical profile during their PM overpass, and its daily maximum during their AM overpass,
which supports the idea that all data collected by A-train instruments (not only CALIPSO)
are very relevant to document the cloud diurnal cycle. This is also roughly the case over land
at altitudes above 8km ASL, although the amplitude of the diurnal variability is quite
underestimated.
In the future, it would be possible to consider CATS measurements at smaller scales, to
identify regionally consistent cloud populations and diurnal behaviors over specific regions
of interest. It would also be possible to use CATS detection of opaque cloud layers to identify
the best local time of observation from space to study local cloud radiative effects. We hope
to address these lines of research in upcoming papers.
**Acknowledgments**
CALIPSO and CloudSat datasets were provided by the NASA Langley Research Atmospheric
Science Data Center (ASDC) and through the AERIS and ICARE/CGTD Data services. Data
were analyzed on the Climserv IPSL computing facilities. This research was made possible
through support by CNRS. We want to thank J.-L. Baray and N. Montoux for useful
discussions.




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



Figures.

Figure 1: (top) Map of the portion of the Earth sampled by CATS, in white: ~78% of the
Earth's surface. (bottom) Evolution of the vertical profile of Cloud Fraction as a function of
local time of observation over the Ocean (left) and Land (right), using CATS detections made
in JJA from 2015 to 2017.

Figure 2. Like Fig. 1, over the North Hemisphere midlatitudes (top row) and Tropics (second
row), the South Hemisphere Tropics (third row) and midlatitudes (bottom row) during JJA
from 2015 to 2017.

Figure 3: Same as Fig. 2, considering data CATS measured during the boreal winter (DJF,
from 2015 to 2017).

Figure 4: The diurnal cycle of clouds as seen (a) by the ground-based LNA lidar from its SIRTA
site in JJA during precipitation-free days over the 2003-2015 period, (b) by the space lidar
CATS in JJA 2015-2017 within a 10°x10° lat-lon box centered on SIRTA, considering only
sunlit conditions for consistency with LNA records, and (c) by the space lidar CATS in JJA
2015-2017 within a 10°x10° lat-lon box centered on the ARM SGP time. Time is local.

Figure 5: Diurnal cycle of the cloud fraction profiles observed by CATS over different
continents a) NH America, b) Europe, c) China, d) NH Africa, e) SH America, f) SH Africa, g)
Australia, averaged over the summer seasons (JJA in the North Hemisphere, DJF in the South
Hemisphere) from 2015 to 2017. For each region we considered all profiles sampled over
land within the boundaries shown by the inset map. CF over Europe do not extend as high
altitudes as the rest, as it is the only region that do not include part of the Tropical band.

Figure 6: Mean diurnal variations of low-level (solid line) and high-level (dotted line) cloud
amounts (%) every 3 hours in fve zonal bands over ocean (left) and land (right) in JJA from
CATS for the period 2015-2017. Plots (a-f) are presented in a similar way as Figure 11 in RS99
for comparison.




Figure 7: Vertical Profiles of Cloud Fraction observed by CALIPSO (full line) and CATS (dashed
line) between ±51° around 0130AM (blue), 0130PM (red) and at all times (black), over ocean
(left) and land (right). Measurements were weighted based on the latitude at which they
were made, to account for the different zonal sampling distributions of both instruments.
CALIOP cloud profiles were built using cloud layers from the CALIPSO v4.10 level 2, 5-km
cloud layer product. Only layers with a Cloud/Aerosol Discrimination score (CAD_Score)
above 0.7 were considered to build the CALIOP profiles, and layers with a
Feature_Type_Score above 5 were considered to build the CATS profiles. For both
instruments, we used JJA observations from 2015 to 2017.

Figure 8: Mean Cloud fraction profiles observed by CATS at the overpass local time of the
sun-synchronous space lidars (CALIPSO and the A-train 01:30UTC, ADM 06:00UTC, Earth-
CARE 02:00UTC) compared to the envelope of the whole cloud fraction profile diurnal cycle
observed by CATS (white), averaged between ±51° over ocean (left) and land (right).
CALIPSO and Earth-CARE are dedicated to clouds an aerosols studies, while ADM is primarily
dedicated to wind measurements in non-cloudy conditions. We used CATS observations
during JJA from 2015 to 2017.





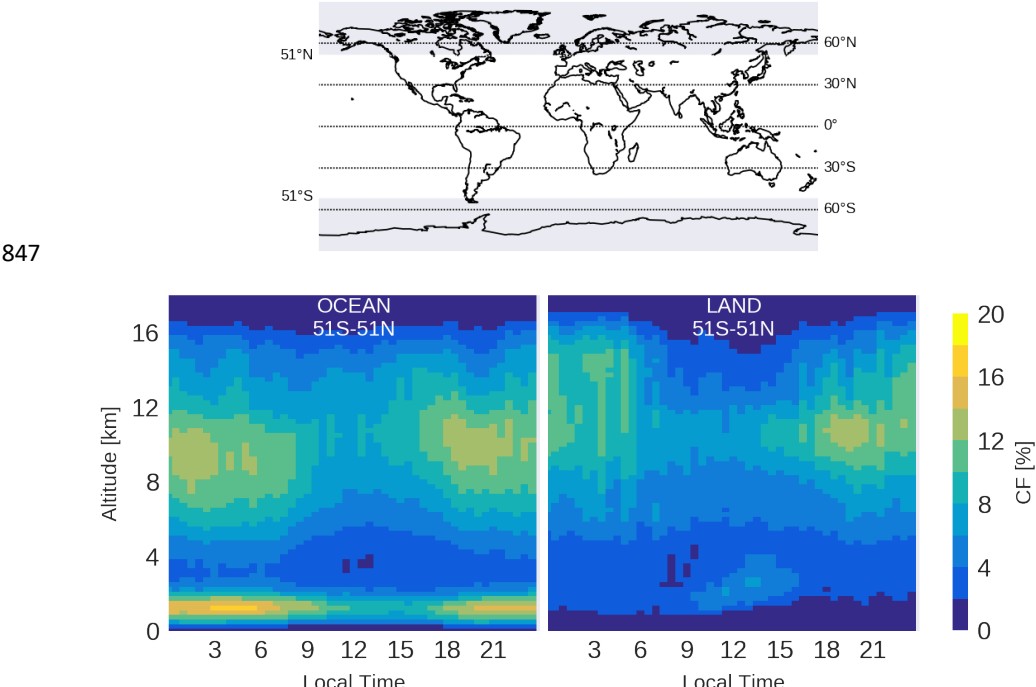


Figure 1: (top) Map of the portion of the Earth sampled by CATS, in white: ~78% of the
Earth's surface. (bottom) Evolution of the vertical profile of Cloud Fraction as a function of
local time of observation over the Ocean (left) and Land (right), using CATS detections made
in JJA from 2015 to 2017.



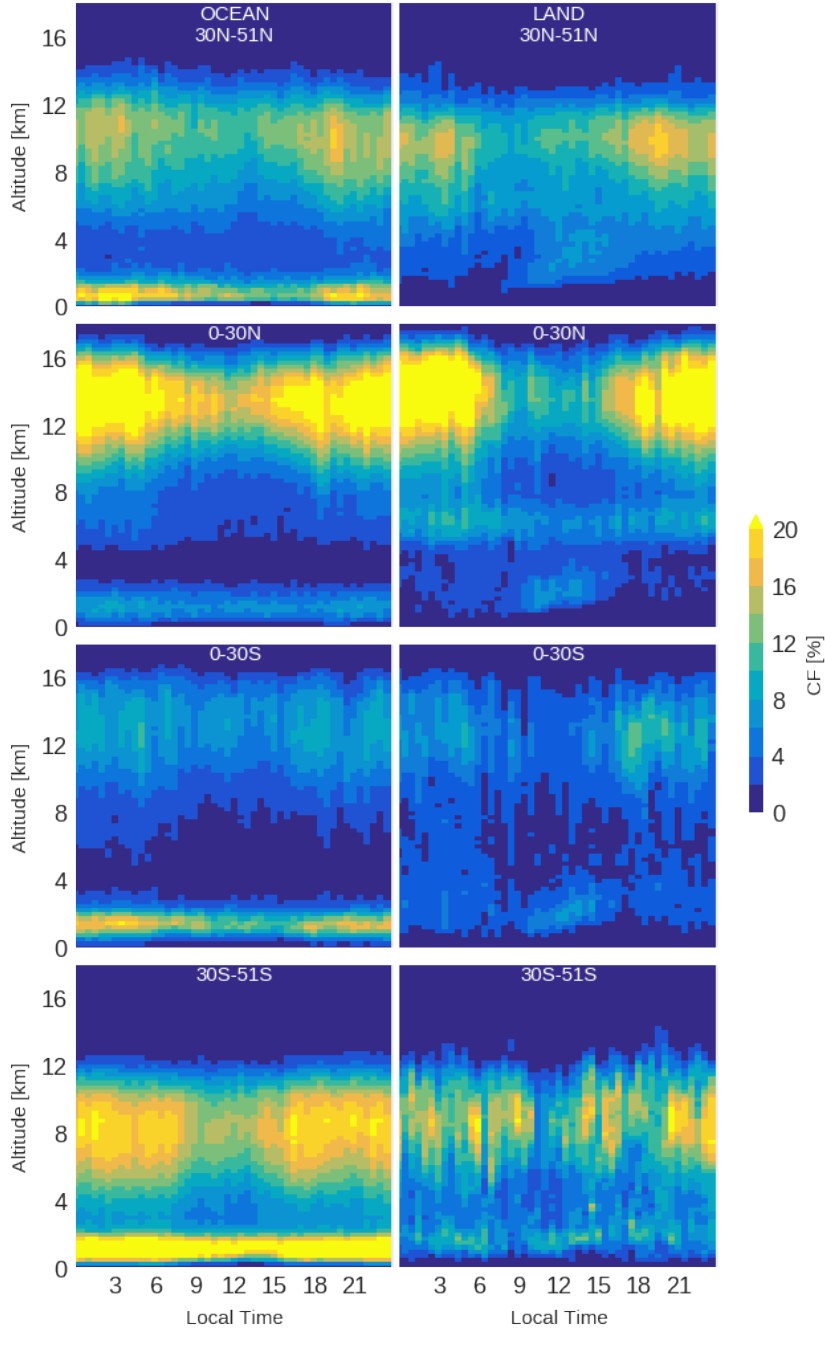


Figure 2. Like Fig. 1, over the North Hemisphere midlatitudes (top row) and Tropics (second
row), the South Hemisphere Tropics (third row) and midlatitudes (bottom row) during JJA
from 2015 to 2017.






Figure 3: Same as Fig. 2, considering data CATS measured during the boreal winter (DJF,

from 2015 to 2017).

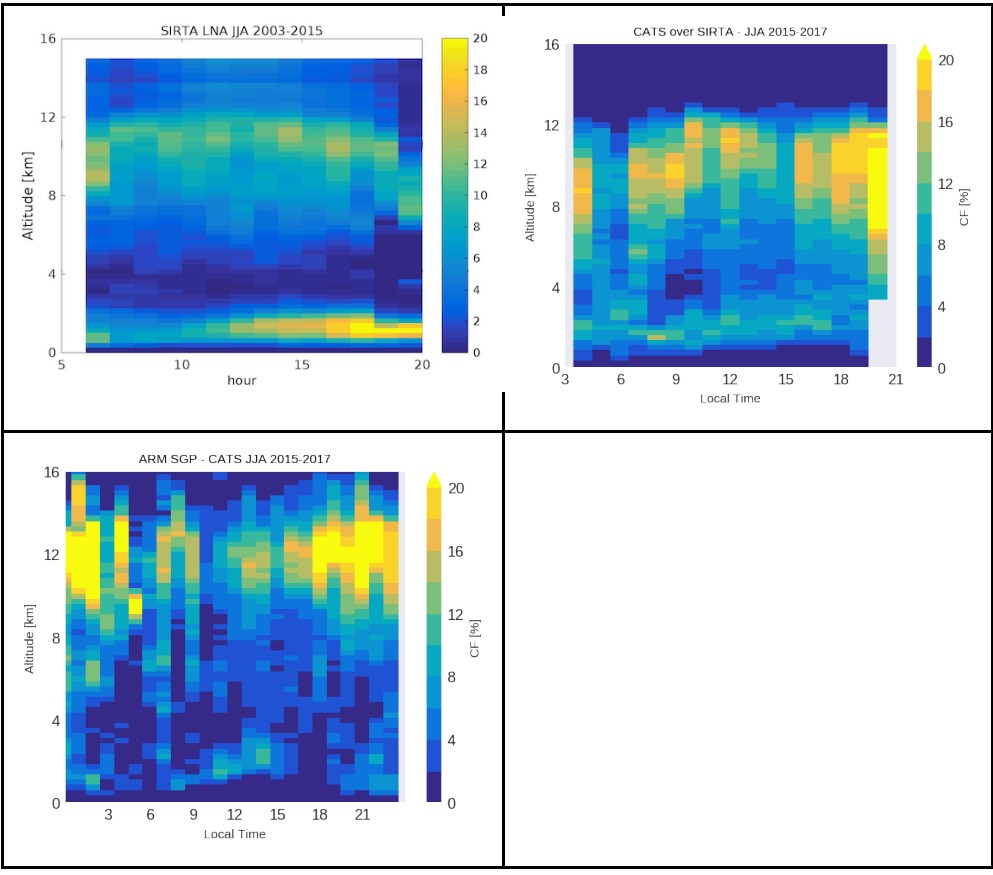


Figure 4: The diurnal cycle cloud profiles as seen (a) by the ground-based LNA lidar from its
SIRTA site in JJA during precipitation-free days over the 2003-2015 period, (b) by the space
lidar CATS in JJA 2015-2017 within a 10°x10° lat-lon box centered on SIRTA, considering only
sunlit conditions for consistency with LNA records, and (c) by the space lidar CATS in JJA
2015-2017 within a 10°x10° lat-lon box centered on the ARM SGP time. Time is local.



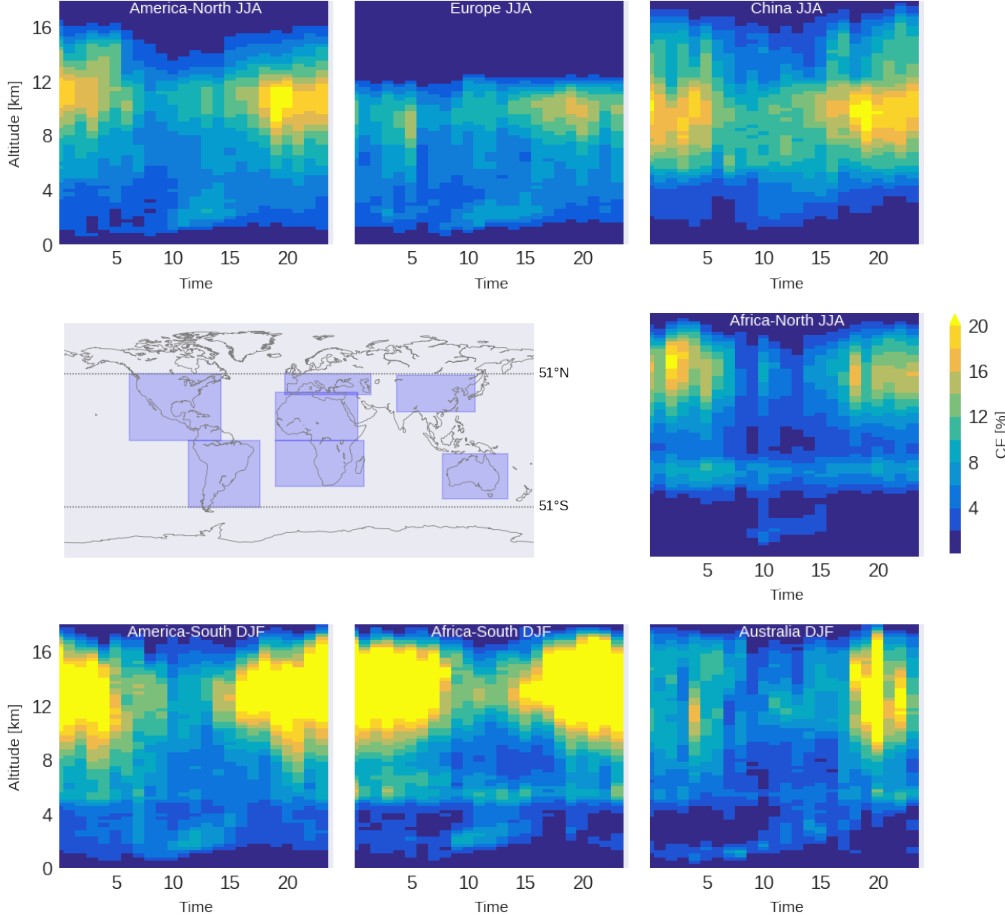


Figure 5: Diurnal cycle of the cloud fraction profiles observed by CATS over different continents a) NH America, b) Europe, c) China, d) NH Africa, e) SH America, f) SH Africa, g) Australia, averaged over the summer seasons (JJA in the North Hemisphere, DJF in the South Hemisphere) from 2015 to 2017. For each region we considered all profiles sampled over land within the boundaries shown by the inset map. CF over Europe do not extend as high altitudes as the rest, as it is the only region that do not include part of the Tropical band.






Figure 6: Mean diurnal variations of low-level (solid line) and high-level (dotted line) cloud
amounts (%) every 3 hours in five zonal bands over ocean (left) and land (right) in JJA from





CATS for the period 2015-2017. Plots (a-f) are presented in a similar way as Figure 11 in RS99
for comparison.




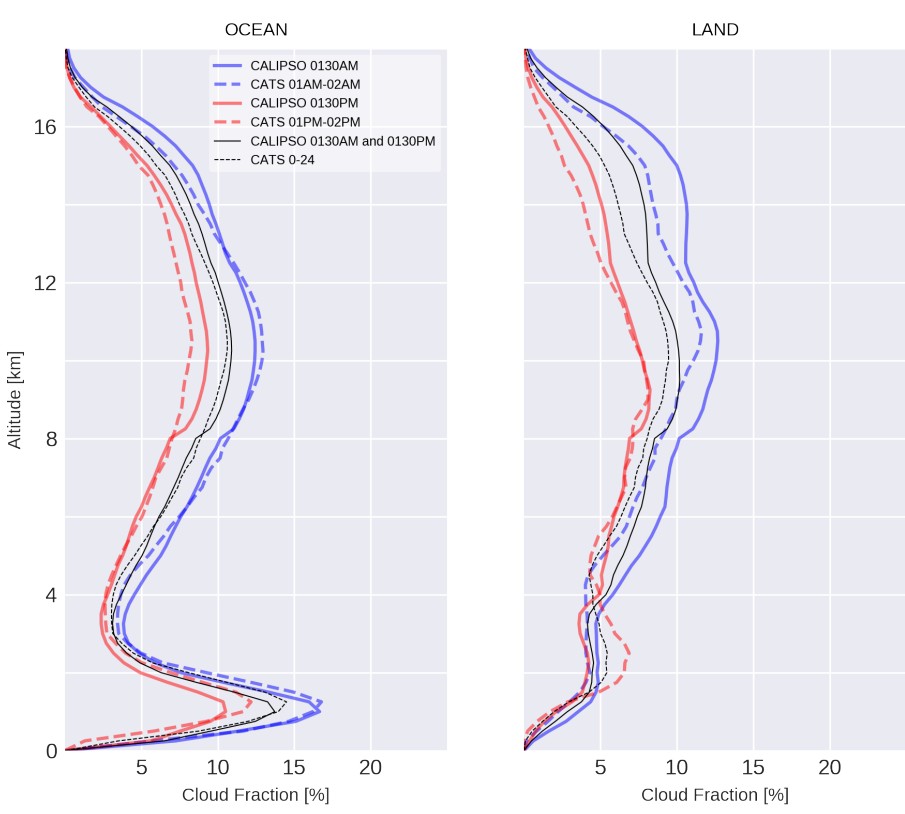


Figure 7: Vertical Profiles of Cloud Fraction observed by CALIPSO (full line) and CATS (dashed
line) between ±51° around 0130AM (blue), 0130PM (red) and at all times (black), over ocean
(left) and land (right). Measurements were weighted based on the latitude at which they
were made, to account for the different zonal sampling distributions of both instruments.
CALIOP cloud profiles were built using cloud layers from the CALIPSO v4.10 level 2, 5-km
cloud layer product. Only layers with a Cloud/Aerosol Discrimination score (CAD_Score)
above 0.7 were considered to build the CALIOP profiles, and layers with a
Feature_Type_Score above 5 were considered to build the CATS profiles. For both
instruments, we used JJA observations from 2015 to 2017.

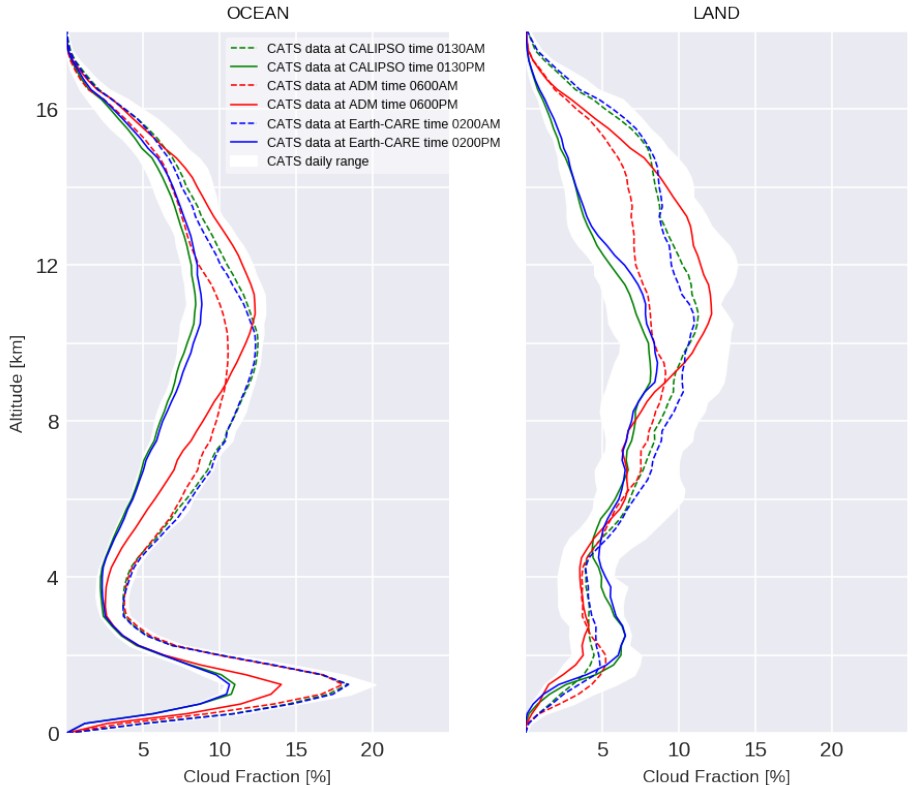

Figure 8: Mean Cloud fraction profiles observed by CATS at the overpass local time of the
sun-synchronous space lidars (CALIPSO and the A-train 01:30UTC, ADM 06:00UTC, Earth-
CARE 02:00UTC) compared to the envelope of the whole cloud fraction profile diurnal cycle
observed by CATS (white), averaged between ±51° over ocean (left) and land (right).
CALIPSO and Earth-CARE are dedicated to clouds an aerosols studies, while ADM is primarily
dedicated to wind measurements in non-cloudy conditions. We used CATS observations
during JJA from 2015 to 2017.



**Appendix A - CATS sampling compared to CALIPSO**

Figure A1 shows the number of profiles sampled by CATS divided by the number of profiles
sampled by CALIOP in the same 2° latitude band, when aggregated over two successive JJA
seasons (2015-2016). The red line considers all CATS profiles, while the green line only
considers CATS profiles sampled roughly around the local time sampled by CALIOP -- i.e. the
green line shows CATS measurements made at the same local time as CALIOP. These results
are based on CALIOP's v4.10 level 2, 5-km cloud layer product and CATS's v2.05 level 2, 5-km
cloud layer product.

The orbital differences between the CALIPSO satellite and the ISS mean that CATS samples
generally less profiles than CALIOP, with a 0.4 minimum ratio near 20°N during the JJA
period. At that latitude, CALIOP samples more than double the number of profiles sampled
by CATS, when considering all local times. When considering only profiles sampled by CATS
at the same local time as CALIOP, the ratio drops to 0.1, meaning CALIOP's sampling is ten
times better the one from CATS. This ratio means that CATS data need to be aggregated over
long periods for any comparison between both instruments to be meaningful.

When considering high latitudes (50° and above), CATS sampling improves significantly, up
to the point where it gets better than CALIOP's: the CATS to CALIOP sampling ratio reaches
1.4 for latitudes above 50°S.



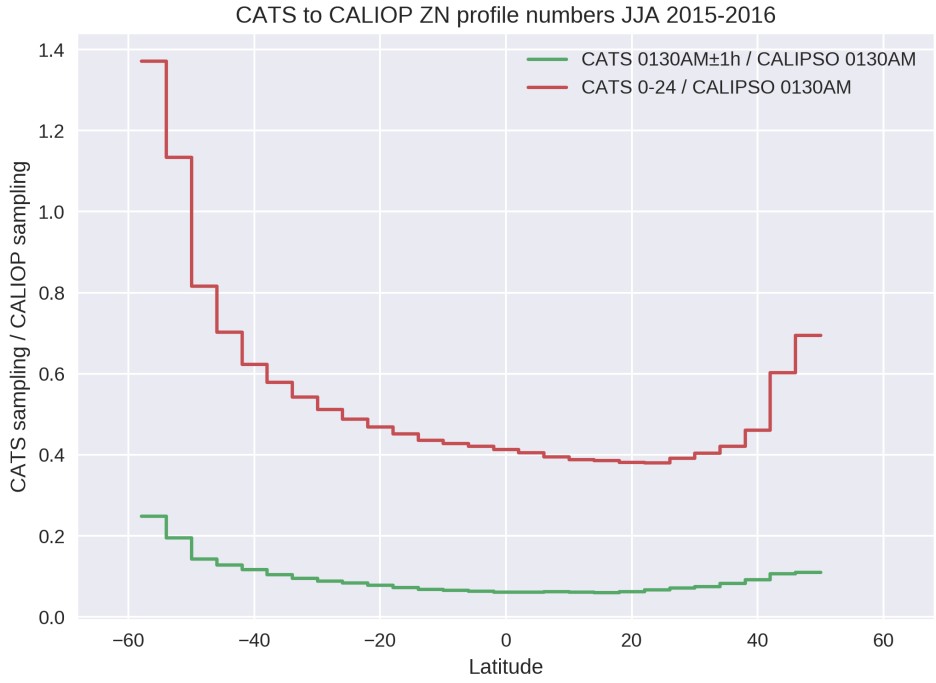


Fig. A1: ratio of the number of profiles seen by CATS and CALIOP in 2° latitude bands over
the JJA periods of 2015 and 2016.



**Appendix B - Continuity of CATS cloud detections according to solar pollution**

Fig. A2 shows Cloud Fraction profiles observed by CATS over land (left) and ocean (right)
between 5 and 8 AM local time in nighttime (orange) and daytime (blue) conditions. Here
we show that cloud detections made using data acquired by CATS either in daytime (sunlit)
conditions (blue) or nighttime conditions (orange) leads to similar cloud fraction profiles.
This suggests that CATS cloud detections are consistent in both conditions and that the
instrument can provide a continuously stable record of cloud detections throughout the day.

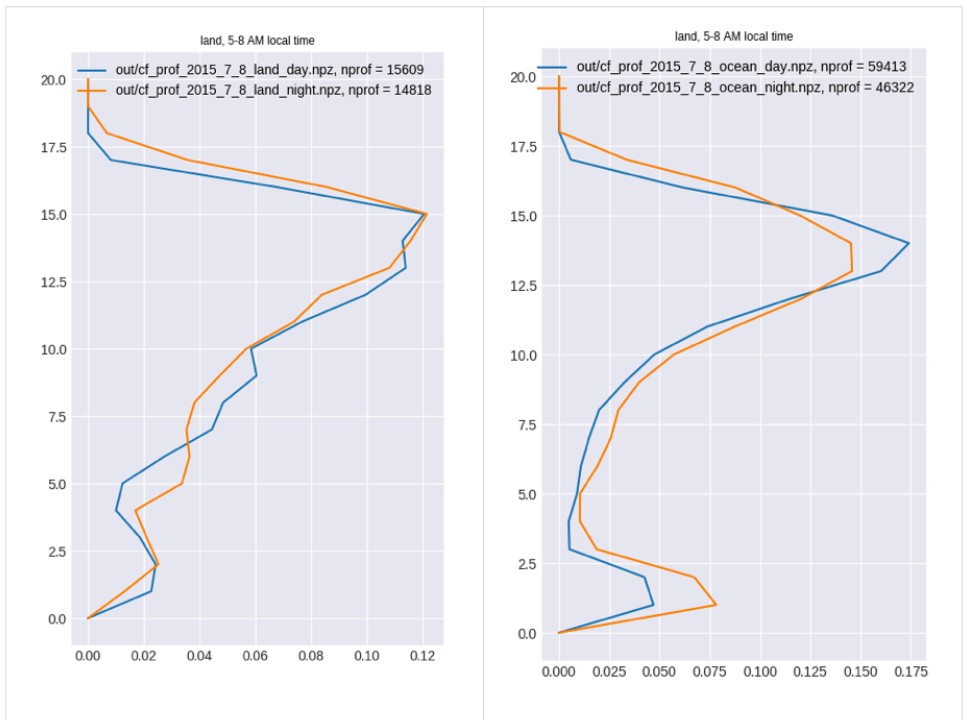


Fig. A2. Profiles of Cloud Fraction observed by CATS over land (left) and ocean (right)
between 5 and 8AM local time (JJA 2015-2016) in nighttime (orange) and daytime (blue)
conditions.

**Appendix C - Sampling bias due to CATS lidar attenuation, by region**

Figures A3 to A7 below document how the CATS sampling get relatively degraded from high
to low altitudes due to the attenuation of the laser light as it gets progressively scattered by
atmospheric components, for the various regions described in the main article. Sampling is
reported relative to its initial value of 100% at high altitudes, where no attenuation has yet
occurred.

Fig. A3 -  Vertical sampling over ocean (left) and land (right) between 51°S and 51°N (same
data as in Fig. 1 in the main article)

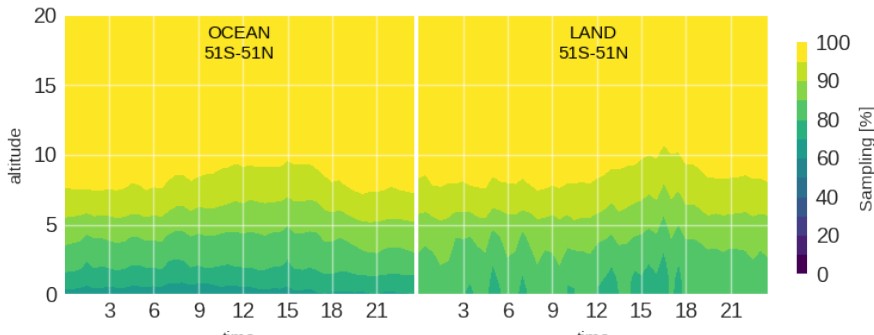








Fig. A4 - Vertical sampling over ocean (left) and land (right) in latitude bands during JJA
(same data as in Fig. 2 in the main article)

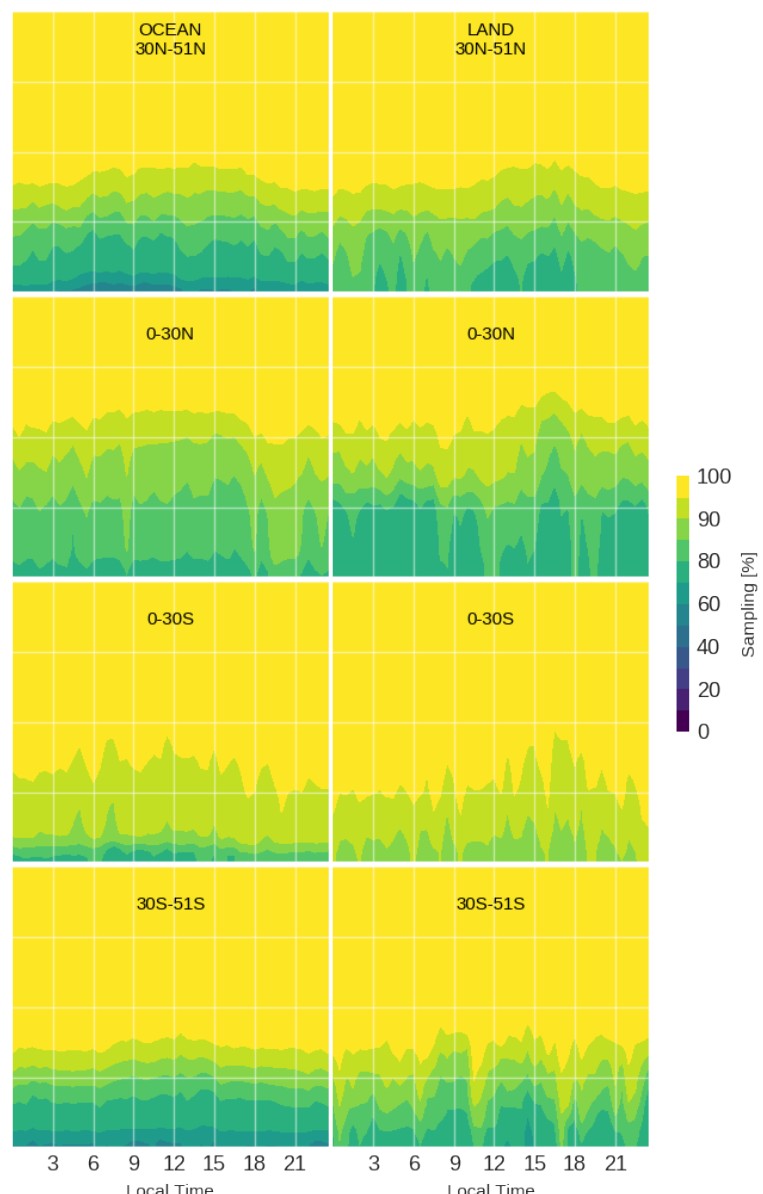







Fig. A5 - Vertical sampling over ocean (left) and land (right) in latitude bands during DJF
(same data as in Fig. 3 in the main article)

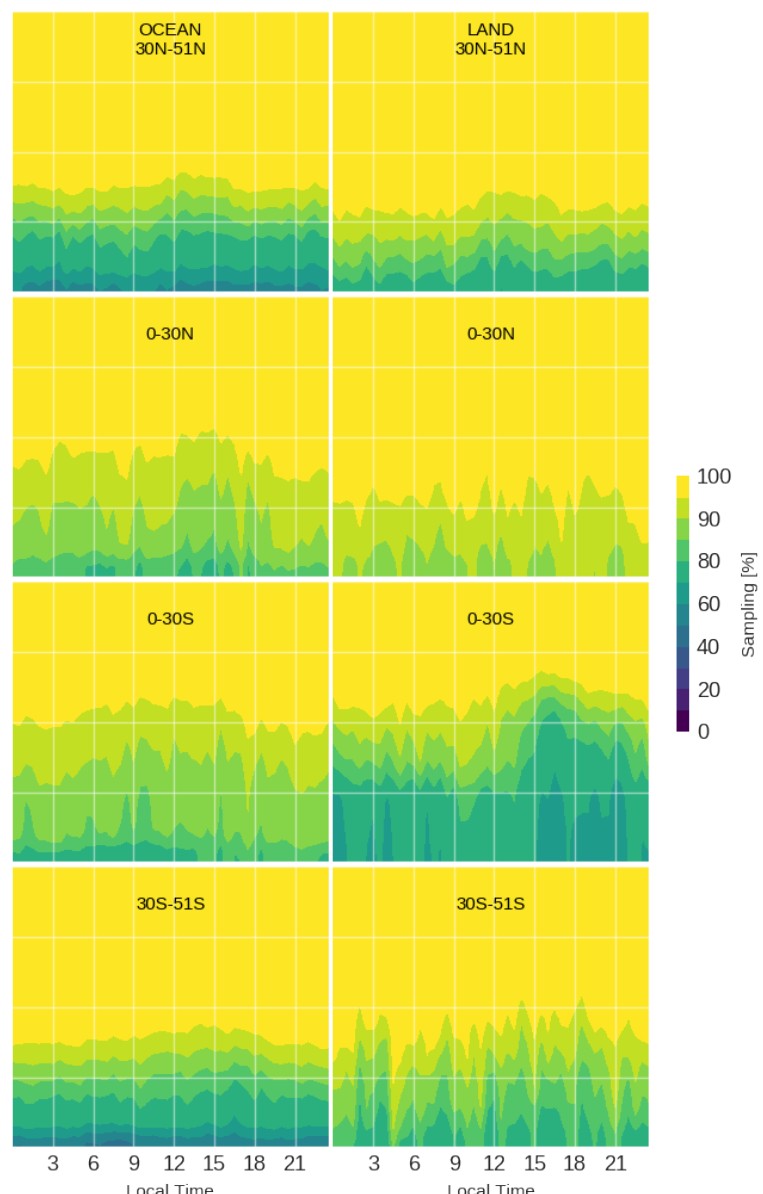








Fig A6 - Vertical sampling of the CATS lidar over the SIRTA ground-based site (top) and over
the ARM SGP site (bottom), same data as in Fig. 4 in the main article

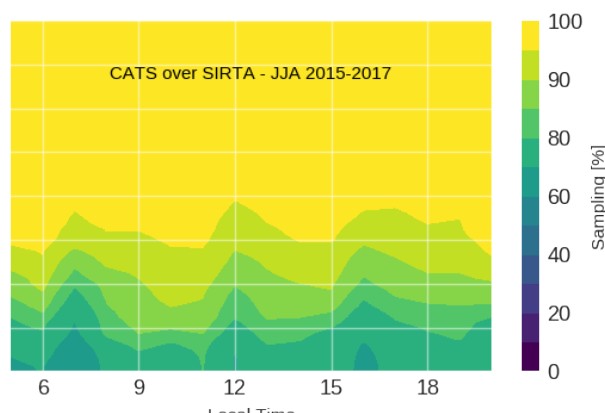


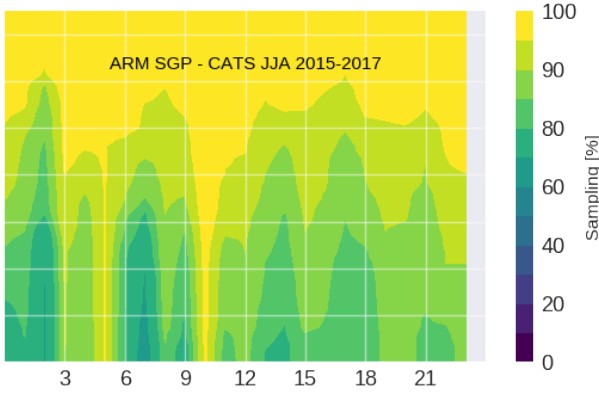








Fig. A7 - Vertical sampling over the regions considered in Fig. 5 from the main paper, during
JJA.

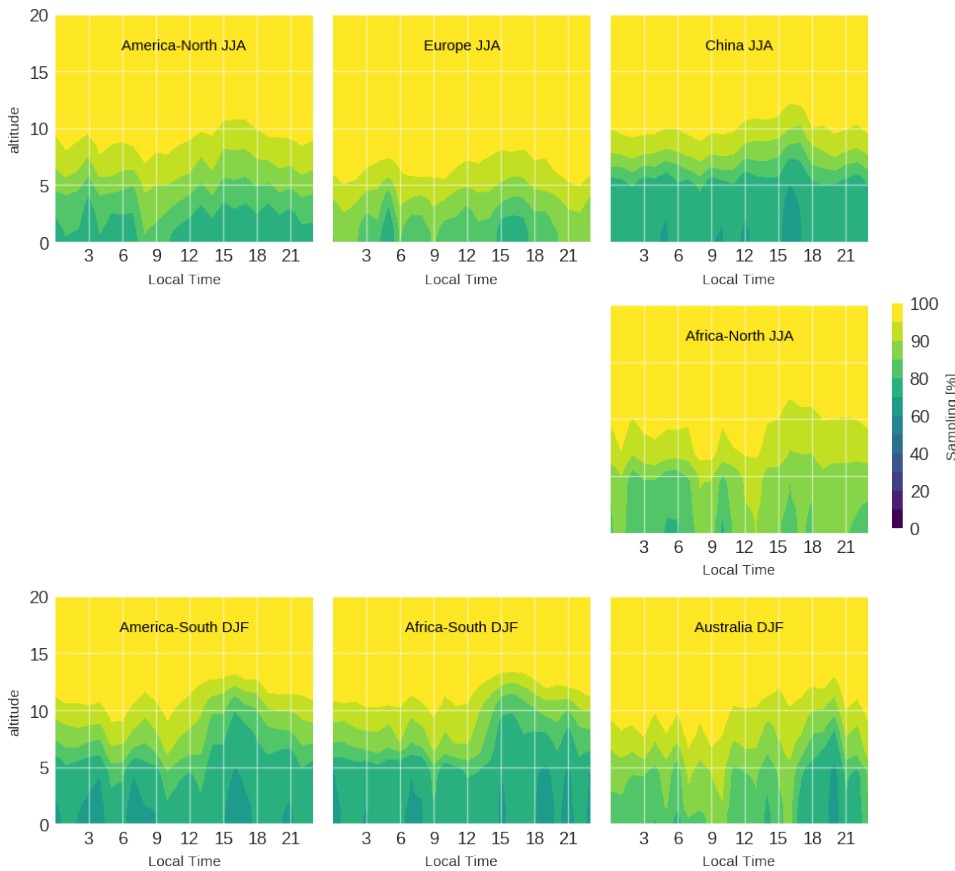

