# Peer review of "The diurnal cycle of cloud profiles over land and ocean between 51°S and 51°N, seen by"

_Atmospheric Chemistry and Physics, 2018_

## Referee Comment (RC1) · Anonymous Referee #1 · 2 Apr 2018

This is a review of "The diurnal cycle of cloud profiles over land and ocean between 51°S and 51°N, seen by the CATS spaceborne lidar from the International Space Station"

This paper presents the cloud detection statistics from the CATS lidar that was operating on the ISS. Because of the non-sun-synchronous orbit of the ISS, these statistics sample all hours of day and night. This creates a unique dataset. This data is presented very well in the paper. I believe this is an excellent paper that will be cited a lot. I certainly recommend publication of the paper in ACP. There are a few minor issues

that I recommend the authors to consider. Those are discussed below.

Like any lidar, CATS probes the first ~3 optical depths of a cloud, as discussed in the paper. In the case of thin cirrus clouds, the full extent of the clouds will be sampled, but in many cases essentially only the top height will be detected. However, the authors confuse this sampled vertical cloud fraction with statistics of vertical extent. For example, the abstract states "the high clouds geometric thickness increases significantly from 1km near 5PM to 5km near 10PM". However, it could also be that the cloud top altitude is more variable later in the day, while the geometrical thickness is staying the same. The data could be analyzed in other ways to include transparent clouds only, which will allow a study of statistics of geometric thickness, but this is not done in the current study. I am not asking to change the study to include this analysis, but the authors should discuss the fact that real geometric extent is not always sampled here. Especially in the tropics a substantial part of the high clouds would be tops of convection that may have vary throughout the day. Other parts of the paper that refer to geometric thickness of clouds are at lines 340-344, 444, 492, 629, 636, and 642. There may be other instances. Please go through the paper and discuss this interpretation of the data correctly.

Line 172: If I understand correctly, lidar depolarization information is used for cloud classification. If so please briefly discuss this in the paper.

Line 372: It seems that a reference to Johnson et al. (1999; J. Climate, 12, 2397–2418) about the tri-modal nature of tropical convection is in place here.

Line 515: Another thing to note is that, besides cloud detection, retrieving a cloud top height from passive instruments is not as straightforward as it is for lidar measurements, especially for thin clouds and in multi-layered situations.

Figure 5 (and A7): I would suggest to add a vertical scale to the Africa-North plot, or maybe to all of the plots.
* * *
Interactive
comment

Figure 6: Because of the ISS orbit, CATS samples between 51 degrees north and south, as explained in the paper. However, figure 6 and the discussion are not consistent with this geographical limitation and include statistics supposedly from latitude bands of 30-60 degrees north and south. This choice is made to be consistent with previous studies, but hides the fact that CATS is only sampling to 51 degrees, making the data not completely consistent with previous datasets. It is important to be consistent about the sampling region throughout the paper. Also, I find the labels of the latitude bands on the right side of figure 6 rather confusing. It makes it seem like vertical axis are latitudes in addition to cloud amount deviation somehow. I would propose adding the latitude bands on top or inside the figure as a label or legend.

---

## Referee Comment (RC2) · Anonymous Referee #2 · 16 Apr 2018

Review "The diurnal cycle of cloud profiles over land and ocean between 51°S and 51°N, seen by the CATS spaceborne lidar from the International Space Station" by Noel et al.

By using CATS measurement, the paper presents a first land-ocean contrast of cloud diurnal cycle. Results are very useful. However, there are many uncertainties associated with CATS data for diurnal cloud studies, which need to be clearly discussed. I suggest the paper for publication after the following comments are properly addressed.

[Figure]

Major issues:

1. There are many challenges in using CATS data to study diurnal cloud cycle. First, it is linked with space lidar observations itself. Although several points (day-night solar background difference, attenuation of lidar signal by upper and middle clouds) are touched in the paper, they are needed to be clearly presented and quantified. Results discussions need to consider these uncertainties.

2. It needs to be very clear that CATS from ISS don't provide exact diurnal cycle cloud observations as ground-based observations. Due to the nature of ISS orbit characters, you need to combine over a month-long measurements together to cover the diurnal cycle. So, composed the diurnal cycle include seasonal cloud variations. Although it is fine to perform the seasonal study as discussed in the paper, it is important to make readers aware of the nature of CATS diurnal cloud properties. Thus, related information needs to be added in the introduction or the method section.

3. One way to make these limitations well understood is by using ground-based observations to validate CATS results. Although there is one figure for this purpose, it is not enough. Tropical observations and over oceans are needed. ARM observations are available for the validations.

Minor issues

1. L23-24: change "high clouds maximum" to "high cloud thickness maximum." The interpretation of cloud thickness detected by a lidar has to consider cloud optical thickness.

2. Line 88-101: Some references are needed her to support the discussion. For example, the Fig. 9 of Wang and Sassen 2001, will support your middle latitude discussion.

Wang, Z., and K. Sassen, 2001: Cloud type and macrophysical property retrieval using multiple remote sensors. J. Appl. Meteor., 40, 1665-1682.

3. Line 106-107: There are many more important related papers should be cited than

your paper.

4. L165: "measured every 350m" not accurate. It is a 350 m average profile.

5. L 171: What is "L2O"?

6. L201-202: So you shouldn't use this site considering it data collection biases.

7. L273-274, "low clouds have their base below 4km ASL": Do you sure that you mean cloud base height here. If so, it does not make sense. First, it is almost impossible for you to detect the base of optically thick clouds. Assuming that you can detect, we refer clouds with the base higher than 2 km as middle-level clouds. Using top height will make more sense.

8. L308-308: Not necessarily true. How often do you detect low clouds below high clouds? Even if high cloud occurrences are high, they are not 100

9. L315-316: Solar-background variations need to be better quantified.

10. L336-346: To what extent, these variations are due to the lower daytime detection sensitivity, especially considering the contrast between N 30-50 with S30-50?

11. L368-374: The high occurrence of middle-level clouds are well documented by may early studies (Zhang et al. 2010; Sassen and Wang 2012, and other), which should be properly referenced.

Zhang, D., Z. Wang, and D. Liu (2010), A global view of midlevel liquid-layer topped stratiform cloud distribution and phase partition from CALIPSO and CloudSat measurements, J. Geophys. Res., 115, D00H13, doi:10.1029/2009JD012143.

Sassen, K. and Z. Wang, 2012: The Clouds of the Middle Troposphere: Composition, Radiative Impact, and Global Distribution, Surv Geophys (2012) 33:677-691,DOI 10.1007/s10712-011-9163-x

12. L411-473: This part of the discussion should occur early in the paper as validation

efforts.

13. L421-422: Considering the night time sampling biases, I don't think that you can trust this result.

14. L449-452: It will good to include a panel for SGP ground-based observation results here.

15. L484-487: In Fig. 5, why cloud top in Europe JJA is significantly lower than the other regions?

16. L522: Where is ISCCP data? Is there any reason not to plot it?

17. L539-541: This could also due to the different day-night cloud detection sensitivities between lidar and ISCCP passive measurements.

18. L574-579: You could try to use CALIOP 1064 only measurements to run the same detection to minimize the difference.

19. L585 "Cloud Fraction": either use CF or "cloud fraction".

20. Section 5: It will good to have some discussion on the potential limitations here.

---

## Short Comment (SC1) · 16 Apr 2018

The paper's discusses the diurnal changes in cloud fraction, but commonly CF is meant to represent the fraction of a grid box area or sensor field of view that is covered in cloud. Would it not be correct to give the results as cloud frequency instead, as that is what is actually being measured?

Secondly, the figures' color bars max out at CF=20%, with values above 20% visible in many of the figures. It would be good to extend the color bar so that fewer figures

saturate like this.

Thanks for the discussion at EGU, this is quite interesting work.

---

## Referee Comment (RC3) · Anonymous Referee #3 · 21 Apr 2018

"The diurnal cycle of cloud profiles over land and ocean between 51S and 51N, seen by the CATS spaceborne lidar from the International Space Station" by Vincent Noel et al.

This paper documents the diurnal cycle of the cloud vertical profiles over a large part of the globe, using CATS lidar, operating on the International Space Station. Cloud fractions from different locations, seasons, instruments have been compared, by taking the advantage of this unique dataset. The study is interesting and useful. But it would be better to relate the role of dynamic and thermodynamic processes to the differences of CF found from different conditions, which is not clearly presented. I recommend some modifications to improve the paper before publication.

Major issues:
1. The second paragraph of Introduction needs more support references to help the readers to better understand the background. For example, 'well documented by passive satellite imagery', it would be better to add in relative works. The same suggestion for 'b) cloud detections from ground-based active instruments' part. There are a lot of works have been done with ground-based instruments on cloud property analysis, I would appreciate if you can give a few references here.
2. The authors chose ARM SGP site for the comparison, is there any particular reason to compare two mid-latitude continental sites with CATS? Comparing with oceanic type of clouds would be interesting, if there are any possibilities. Since you actually didn't really process the SGP dataset, I would suggest that you could keep this part as an additional material.
3. You talked about the pronounced mid-level clouds over continent many times which has been well documented, could you give a more detailed explanation for that, and the role of dynamic and thermodynamic processes.

Minor issues:
1. On Figure 1, is there any way to show the number of samples on the plots as well? In the text, the latitude range is 51S-51N, but on most of the plots, it's 55S-55N. It is better to keep it consistent.
2. The color bar need to be adjusted and extended to greater than 20%, add in unit, and keep the x axis and y axis consistent for the same figure group, specially figure 4.
3. Figure 5, better to label A, B, C… on the subplot for each location.
4. Line 515: Using passive instruments to retrieve the cloud properties is different from active instruments, they don't have the same sensitivity for the thin clouds. It isn't a fair comparison here.
5. Another thing to note is that, besides cloud detection, retrieving a cloud top height from especially for thin clouds and in multi-layered situations.

---

## Author Comment (AC1) · 15 May 2018

Thanks for your review. Regarding your comment:

*One way to make these limitations well understood is by using ground-based observations to validate CATS results. Although there is one figure for this purpose, it is not enough. Tropical observations and over oceans are needed. ARM observations are available for the validations.*

We have tried locating a well-documented, 24/24 dataset of cloud layers covering the

period 2015-2017 based on measurements from a ground-based lidar operating in the Tropics, preferably close to the ocean. We have contacted several observatories (e.g OPAR) but it appears cloud layer products are often unvalidated and/or suffer from irregular or non-diurnal sampling. Deriving robust cloud statistics based on those, while possible, is an involved and long process that often requires its own study, and we could not use those. We understand that validation of CATS detections through comparison with an external dataset would in theory require that external dataset to be validated itself, perhaps by being used in a published article.

We investigated ARM data from https://www.arm.gov/data and found several datasets based on Tropics measurements and promising cloud layer information. We found that

- Datasets from Nauru Island and Darwin Australia did not overlap with CATS timeframe

- Datasets from Brazil and Ascension Island cover the CATS timeframe but only contained profiles of Attenuated Backscatter (without cloud detection) - doing the cloud detection ourselves would require validating the obtained cloud dataset first (as explained above)

- Only datasets from the ARM Eastern North Atlantic (ENA) atmopheric observatory - https://www.arm.gov/capabilities/observatories/ena - are close to our criterias above. This site provides cloud layers derived from ground-based lidar measurements made in an oceanic environment, unlike the SIRTA and ARM-SGP datasets considered in the initial article. Our initial exploration of the enaarsclka-zrbnd1kolliasC1 dataset (based on a combination of lidar and radar data) showed problems during the 2017 summer due to issues with lidar cloud detections. We contacted ARM people, who suggested rebuilding the cloud layers based on the cloud mask source product and ignoring the lidar-only detections, which resolved the problem but in effect turned it into a radar-based product. Moreover, the ENA observatory is located at 39°N, i.e. its latitude is too high to make it Tropical.
In conclusion, our search for ground-based lidar observations has so far only turned up a single dataset that is located outside the Tropics and suffers from instrumental issues. So far we have been unable to locate an appropriate dataset.

Would you consider the ARM-ENA cloud mask dataset to provide a satisfactory basis for CATS comparison? If not, could you suggest a more appropriate and robust dataset?

---

## Author Response (AR1)

**The diurnal cycle of cloud profiles over land and ocean between 51°S and 51°N, seen by the CATS spaceborne lidar from the International Space Station**

Vincent Noel [1], Hélène Chepfer [2], Marjolaine Chiriaco [3], John Yorks

**Reply to Reviewers**
June 19th, 2018

Original reviewer comments are in blue italics, our replies in black.

**Reviewer 1 comments and replies**

*This is a review of "The diurnal cycle of cloud profiles over land and ocean between 51◦S and 51◦N, seen by the CATS spaceborne lidar from the International Space Station"*

*This paper presents the cloud detection statistics from the CATS lidar that was operating on the ISS. Because of the non-sun-synchronous orbit of the ISS, these statistics sample all hours of day and night. This creates a unique dataset. This data is presented very well in the paper. I believe this is an excellent paper that will be cited a lot. I certainly recommend publication of the paper in ACP. There are a few minor issues that I recommend the authors to consider. Those are discussed below.*

*Like any lidar, CATS probes the first ∼3 optical depths of a cloud, as discussed in the paper. In the case of thin cirrus clouds, the full extent of the clouds will be sampled, but in many cases essentially only the top height will be detected. However, the authors confuse this sampled vertical cloud fraction with statistics of vertical extent. For example, the abstract states "the high clouds geometric thickness increases significantly from 1km near 5PM to 5km near 10PM". However, it could also be that the cloud top altitude is more variable later in the day, while the geometrical thickness is staying the same. The data could be analyzed in other ways to include transparent clouds only, which will allow a study of statistics of geometric thickness, but this is not done in the current study. I am not asking to change the study to include this analysis, but the authors should discuss the fact that real geometric extent is not always sampled here. Especially in the tropics a substantial part of the high clouds would be tops of convection that may have vary throughout the day. Other parts of the paper that refer to geometric thickness of clouds are at lines 340-344, 444, 492, 629, 636, and 642. There may be other instances. Please go through the paper and discuss this interpretation of the data correctly.*

This interpretation is correct, and we thank the reviewer for pointing out this problem. We went through the paper (thanks for the pointers) and now try to present the reader a more correct interpretation of the results.

*Line 172: If I understand correctly, lidar depolarization information is used for cloud classification. If so please briefly discuss this in the paper.*

A sentence has been added to the CATS overview paragraph that briefly outlines the CATS cloud phase algorithm and references the appropriate papers for more details. The full description of the CATS cloud phase algorithm is presented in Section 4.3 of the CATS ATBD [1] and in an AMT paper soon to be submitted. High confidence liquid water clouds are classified if the cloud layer has a $T_{mid} > 0$ C and high confidence ice clouds are identified as cloud layers with a $T_{mid} < -20$ C. These ice clouds and liquid water clouds are assigned a CP score of 10 and -10, respectively. Next, the CP algorithm identifies high confidence ice cloud layers as those layers with 1064 nm depolarization ratios greater than 0.25 or $T_{mid} < -10$ C (CP Score = 9). High confidence liquid water clouds are classified if the cloud layer has a 1064 nm depolarization ratio< 0.15 (CP Score = -9). The remaining layers are determined to have lower confidence cloud phase and are assigned a CP Score with an absolute value of 7 or less. These thresholds are based on Yorks et al. (2011) and Hu et al. (2009). Comparisons with CALIOP have shown very good agreement between the two instruments for cloud phase.

[1] https://cats.gsfc.nasa.gov/media/docs/CATS_ATBD_V1-02.pdf

*Line 372: It seems that a reference to Johnson et al. (1999; J. Climate, 12, 2397–2418) about the tri-modal nature of tropical convection is in place here.*

We thank the reviewer for this very useful reference and comment, which are now both included in the text. Note however that CATS only reports a significant population of those midlevel clouds (5-7km) over land, and not over ocean. This is not consistent with those clouds being cumulus congestus, as these also appear over ocean (Masugana et al. 2005). Higher-altitude clouds, which cloud mask such clouds from the lidar view, are equally frequent over ocean and land, so this inconsistency is not explained by instrumental bias.

Following the references provided by Reviewer 2 suggests those clouds could be Altocumulus, as both share middle-level altitudes and locations over land in the summer hemisphere. This possibility is also now mentioned in the text.

*Line 515: Another thing to note is that, besides cloud detection, retrieving a cloud top height from passive instruments is not as straightforward as it is for lidar measurements, especially for thin clouds and in multi-layered situations.*

We agree with the reviewer and have modified the text to include this point.

*Figure 5 (and A7): I would suggest to add a vertical scale to the Africa-North plot, or maybe to all of the plots.*

Following this comment, we have added vertical scales to all the subplots of Figures 5 and A7.

*Figure 6: Because of the ISS orbit, CATS samples between 51 degrees north and south, as explained in the paper. However, figure 6 and the discussion are not consistent with this geographical limitation and include statistics supposedly from latitude bands of 30-60 degrees north and south. This choice is made to be consistent with previous studies, but hides the fact that CATS is only sampling to 51 degrees, making the data not completely consistent with previous datasets. It is important to be consistent about the sampling region throughout the paper. Also, I find the labels of the latitude bands on the right side of figure 6 rather confusing. It makes it seem like vertical axis are latitudes in addition to cloud amount deviation somehow. I would propose adding the latitude bands on top or inside the figure as a label or legend.*

We agree with the reviewer's position, and have updated figure 6 to be hopefully less confusing, and convey the actual sampling latitude range of CATS.

**Reviewer 2 comments and replies**

*Review "The diurnal cycle of cloud profiles over land and ocean between 51◦S and 51◦N, seen by the CATS spaceborne lidar from the International Space Station" by Noel et al.*
*By using CATS measurement, the paper presents a first land-ocean contrast of cloud diurnal cycle. Results are very useful. However, there are many uncertainties associ- ated with CATS data for diurnal cloud studies, which need to be clearly discussed. I suggest the paper for publication after the following comments are properly addressed.*

**Major issues**

*1. There are many challenges in using CATS data to study diurnal cloud cycle. First, it is linked with space lidar observations itself. Although several points (day-night solar background difference, attenuation of lidar signal by upper and middle clouds) are touched in the paper, they are needed to be clearly presented and quantified. Results discussions need to consider these uncertainties.*

The paper now includes more extensive comparisons with ground-based datasets, that we hope will help the reader understand the strengths and limitations of spaceborne lidar measurements, including the influence of attenuation by upper and middle clouds on the detected low-altitude clouds.

Regarding the day-night variation, the CATS minimum detectable backscatter (MDB) at 1064nm goes from 5.10-5 km-1 sr-1 in absence of sunlight to 1.30 10-3km-1 sr-1 in illuminated conditions (Yorks et al., 2016). CATS daytime profiles are horizontally averaged across 60km before cloud detection, which bring the daytime MDB down to nighttime levels. This has two implications for daytime data: 1) optically thinnest clouds detected during nighttime at 60km horizontal averaging might be absent from daytime detections, these represent roughly ~5% of nightime clouds. 2) cloud amounts might be overestimated when many clouds with small horizontal extent are present - this mainly concerns boundary layer clouds. In our evaluations, the associated decrease in SNR due to solar background has a bigger impact on aerosol layer detection than clouds.

CATS's MDB is smaller than CALIOP's 532nm daytime MDB (1.70 10-3 km-1 sr-1), so all other things being equal CATS should detect more clouds than CALIOP in daytime conditions. Both Sassen et al. (2009) and Gupta et al. (2018) successfully used CALIPSO cloud detections in both nighttime and (solar-affected) daytime conditions to document part of the diurnal variability of clouds and, like us, report more high clouds in nighttime measurements. They remark that the observed nighttime increase is considerably more than the uncertainty that might arise from the daytime loss of detection sensitivity. Since CATS cloud detection abilities are at least on par with CALIOP's in daytime conditions, and CALIOP daytime detections are found acceptable to document part of the diurnal cycle, it follows that the existence of solar pollution in the CATS dataset should not prevent its use to document the diurnal cycle of clouds. As in the Sassen and Gupta papers, we note that how much CALIPSO (and therefore CATS) daytime detections underestimate high clouds occurrence and overestimate low clouds need to be quantified.

These points are now made in the text (Sect. 2.1, 3.1 and 5).

In a similar way, how extinction from high clouds affects the retrieval of low-level clouds remains unquantified and hard to evaluate for all spaceborne lidars. Comparisons with ground-based datasets (see major point 3) suggest that high clouds do not impair significantly the retrieval of low clouds over any site. Over ARM-ENA (oceanic site), the limited amount of high clouds means CATS reports of low clouds amounts is very close to the ground-based one. Over ARM-SGP the relatively large amount of high clouds at night might explain why CATS misses half of the nighttime low and mid-level clouds observed by the ground radar. Supposing that 50% of unsampled profiles due to masking by high clouds are indeed cloudy (i.e. an hypothesis of random overlap) is not sufficient to fix the space-ground disagreement. We think extending these results to the global scale for CATS (and CALIPSO) would be a interesting future project.

The upcoming paper by Yorks et al. (In preparation for AMT) will quantify these points further. We have tried to discuss the importance of uncertainties on the results presented here in the last section of the article.

*2. It needs to be very clear that CATS from ISS don't provide exact diurnal cycle cloud observations as ground-based observations. Due to the nature of ISS orbit characters, you need to combine over a month-long measurements together to cover the diurnal cycle. So, composed the diurnal cycle include seasonal cloud variations. Although it is fine to perform the seasonal study as discussed in the paper, it is important to make readers aware of the nature of CATS diurnal cloud properties. Thus, related information needs to be added in the introduction or the method section.*

Following this comment, we have updated the introduction (before-last paragraph) to make clear 1) that the CATS lidar cannot track the evolution of cloudiness above a particular location along a particular day and 2) that we recreate the cloud diurnal cycle over a given location by aggregating over seasons the cloud detections made by CATS over that particular location at different times of day. We now make this point again in the relevant Data and Methods section (Sect. 2.2.a, 2$^{nd}$ paragraph).

*3. One way to make these limitations well understood is by using ground-based observations to validate CATS results. Although there is one figure for this purpose, it is not enough. Tropical observations and over oceans are needed. ARM observations are available for the validations.*

Following this comment, and major comment #2 from Reviewer 3, we have tried locating a well-documented, 24/24 dataset of cloud layers covering the period 2015-2017 based on measurements from a ground-based lidar operating in the Tropics, preferably close to the ocean. We have contacted several observatories (e.g OPAR) but it appears lidar-based cloud layer products are often unvalidated and/or suffer from irregular or non-diurnal sampling.

Following the Reviewer's suggestion, we investigated ARM data [1] and found several datasets based on Tropics measurements and promising cloud layer information. We found that:
- Datasets from Nauru Island and Darwin Australia did not overlap with CATS timeframe
- Datasets from Brazil and Ascension Island cover the CATS timeframe but only contained profiles of Attenuated Backscatter (without cloud detection) — doing the cloud detection ourselves would require external validation
- Only datasets from the ARM Eastern North Atlantic (ENA) atmospheric observatory [2] are close to our criterias above. This site provides cloud layers derived from ground-based lidar measurements made in an oceanic environment, unlike the SIRTA and ARM-SGP datasets considered in the initial article.

Since the ENA observatory is located at 39°N, it is at best sub-tropical. It is however the only oceanic ARM site we found that could provide a 24/24 robust dataset of cloud layers covering the CATS time period.

Our initial exploration of the enaarsclkazrbnd1kolliasC1 dataset (based on a combination of lidar and radar data) showed unusual results during the 2017 summer (see figure below). We contacted ARM people, who explained the problem comes from unresolved issues with lidar cloud detections and suggested rebuilding the cloud layers based on the cloud mask source product and ignoring the lidar-only detections. This resolved the problem, but in effect turned it into a radar-based product.

[Figure]

*Figure 1 - Cloud fraction over ARM-ENA for 2017 JJA using both lidar and radar cloud detections (left) and radar-only cloud detections (right)*

The paper now includes (Sect. 3.3) a direct comparison of the diurnal cycles of cloud fraction profiles as documented by CATS over the ENA site and from ground-based radar detections. We have also obtained the ARM-SGP ground-based lidar+radar cloud detections in order to directly include them in the paper for comparison with CATS data. Exploration of the sgparsclkazrbnd1kolliasC1 dataset showed artefacts similar to those from the ARM-ENA datasets (vertical steps in cloud fraction) when including lidar-only cloud detections. We thus had again to consider only radar detections, leading to results very similar to those presented in Zhao et al. (2016).

The fact that we uncovered issues with those ARM datasets, which are apparently among the most reliable, confirms how difficult it is to obtain faultless lidar-based cloud retrievals over long periods. Providing our own analysis of the data allowed us to only use cloud detections made during the CATS operation period, making time periods consistent across all ground-based comparisons.

The article now includes comparisons between CATS and:
- Lidar-based cloud retrievals from a midlatitude continental site (SIRTA), already described in e.g. Noel et Haeffelin (2006)
- Radar-based cloud retrievals from a midlatitude continental site (ARM-SGP), already described in e.g. Zhao et al. (2016)
- Radar-based cloud retrivals from a subtropical oceanic site (ARM-ENA).

We hope these improvements to our comparisons between CATS and ground-based datasets better highlights the strenghts and limitations of spaceborne and ground-based lidar cloud sampling.

[1] https://www.arm.gov/data
[2] https://www.arm.gov/capabilities/observatories/ena

**Minor issues**

*1. L23-24: change "high clouds maximum" to "high cloud thickness maximum." The interpretation of cloud thickness detected by a lidar has to consider cloud optical thickness.*

Following the first comment from reviewer 1, the text referenced here has been modified. We hope it now satisfies the concern expressed here.

*2. Line 88-101: Some references are needed her to support the discussion. For example, the Fig. 9 of Wang and Sassen 2001, will support your middle latitude discussion.*

*Wang, Z., and K. Sassen, 2001: Cloud type and macrophysical property retrieval using multiple remote sensors. J. Appl. Meteor., 40, 1665-1682.*

Our initial idea was that paragraph would sum up the findings of the articles referenced in the previous paragraph ("Those studies..."). However, the first paragraph only references studies of cloud diurnal cycles documented from space, whereas the findings in the second paragraph are general. Following this observation, we have revised the first two paragraphs of the introduction, trying to support our assertions with appropriate references, including the one suggested by the reviewer. We thank the reviewer for that very on-point reference.

This comment echoes Major Issue #1 from reviewer 3.

*3. Line 106-107: There are many more important related papers should be cited than your paper.*

The thank the reviewer for pointing this out. We have updated the manuscript to include references to papers that are hopefully more important.

*4. L165: "measured every 350m" not accurate. It is a 350 m average profile.*

The Reviewer is correct. CALIPSO sends every $1/20^{th}$ of a second a laser pulse, which travels to the Earth's surface and back before the satellite has time to move significantly and thus can be considered instantaneous. Unlike CALIPSO, CATS has a high repetition rate of 5kHz and monitors constantly for backscattered energy. The onboard data system accumulates 250 of these 5kHz profiles and reports the data every ±350m to approximate a 20Hz measurement rate.

The text has been changed to "CATS reports vertical profiles of Attenuated Total Backscatter (ATB) every 350m at 1064nm with a 60m vertical resolution (Yorks et al., 2016a). Each profile is created by accumulating backscattered energy from 250 5kHz pulses, 20 times per second."

*5. L 171: What is "L2O"?*

Files for CATS level 2 layer products share the prefix "CATS-ISS_L2O_N-M7.2-V2-01_05kmLay". The L2O designation identifies "Level 2 Operational" products. The updated text now includes this explanation.

*6. L201-202: So you shouldn't use this site considering it data collection biases.*

Unfortunately, the number of site providing datasets containing 24-hour retrievals of cloud boundaries derived from active measurements and part of published research is currently limited, as we explained in our answer to major comment #3. Even lidar-based cloud retrievals from ARM sites suffer from artefacts that prevent their use in this study.

In the updated manuscript we now include, in addition to direct comparisons of CATS with SIRTA lidar retrievals (over midlatitude western Europe), comparisons of cloud retrievals from CATS with others based on measurements from ARM-ENA (subtropical oceanic) and ARM-SGP (midlatitude US) observation sites. As the ARM retrievals are radar-only, we decided to keep the SIRTA dataset as it is the only lidar-based ground-based cloud retrievals dataset.

*7. L273-274, "low clouds have their base below 4km ASL": Do you sure that you mean cloud base height here. If so, it does not make sense. First, it is almost impossible for you to detect the base of optically thick clouds. Assuming that you can detect, we refer clouds with the base higher than 2 km as middle-level clouds. Using top height will make more sense.*

The Reviewer is correct, the original sentence mixed up cloud base and top. Thanks for noticing that error. The text now includes the correct explanation: low clouds have their top below 4km ASL, high clouds have their base above 7km, and mid-level clouds are in between.

*8. L308-308: Not necessarily true. How often do you detect low clouds below high clouds? Even if high cloud occurrences are high, they are not 100.*

The reviewer is correct, the logic of the discussion was incorrect. We have modified the text to fix the discussion and hopefully better make the point we were trying to make.

*9. L315-316: Solar-background variations need to be better quantified.*

We adressed the solar background variations issue in our answer to major comment #1.

*10. L336-346: To what extent, these variations are due to the lower daytime detection sensitivity, especially considering the contrast between N30-50 with S30-50?*

We adressed the issue of solar background variations in our reply to major comment #1.

*11. L368-374: The high occurrence of middle-level clouds are well documented by may early studies (Zhang et al. 2010; Sassen and Wang 2012, and other), which should be properly referenced.*

*Zhang, D., Z. Wang, and D. Liu (2010), A global view of midlevel liquid-layer topped stratiform cloud distribution and phase partition from CALIPSO and CloudSat measurements, J. Geophys. Res., 115, D00H13, doi:10.1029/2009JD012143.*
*Sassen, K. and Z. Wang, 2012: The Clouds of the Middle Troposphere: Composition, Radiative Impact, and Global Distribution, Surv Geophys (2012) 33:677-691,DOI 10.1007/s10712-011-9163-x*

The mid-level clouds CATS detects over Africa, South America and Australia, in the North hemisphere in JJA and the South hemisphere in DJF, might very well be Altocumulus clouds that Wang and Sassen (2012) document in the same locations, altitudes and times. This possibility is now discussed in the text. We thank the Reviewer for this useful reference.

*12. L411-473: This part of the discussion should occur early in the paper as validation efforts.*

We present the results at global scale first on purpose, as we consider those are the most novel and interesting for potential readers. Since validation efforts are not the main purpose of the paper, we think nothing is lost by delaying their presentation.

*13. L421-422: Considering the night time sampling biases, I don't think that you can trust this result.*

We have modified the text to include this observation.

*14. L449-452: It will good to include a panel for SGP ground-based observation results here.*

As noted before, for the revision we have obtained the ARM-SGP ground-based lidar+radar cloud detections from the ARM site. From those, we have extracted radar-only detections (to avoid bias from spurious lidar detections) and derived the daily cycle of vertical cloud fraction profiles during the CATS period (JJA 2015-2017). These results are now included in the paper's Figure 4. They are very similar to those presented in Zhao et al. (2016).

*15. L484-487: In Fig. 5, why cloud top in Europe JJA is significantly lower than the other regions?*

Europe is the only region in Fig. 5 that does not include part of the Tropical band, where the tropopause reaches much higher altitudes. This leads to cloud tops over Europe

We included this information in the legend of Figure 5 in the updated manuscript version that we submitted in March (this version has "over land and ocean" in the title), which superseded our original submission in February and became the one available from the ACPD website during the open discussion [1]. We do not know why the reviewers were provided with the non-updated version and regret the confusion.

[1] https://www.atmos-chem-phys-discuss.net/acp-2018-214/acp-2018-214.pdf

*16. L522: Where is ISCCP data? Is there any reason not to plot it?*

Indeed, we have decided against directly including retrievals based on ISCCP in the paper. By doing so, our goal is to prevent the discussion from focusing on active-vs-passive detection differences, and spending too much time explaining why this instrument detects that much more high clouds here and that much less low clouds there. Those questions are valid, and require thorough discussions about the subtle interplay between instrumental sensitivities, the distribution of cloud properties on a global scale, and data analysis algorithmic choices, all of which require extensive studies of their own (e.g. Stubenrauch et al. 2012 which is 176 pages long). We were concerned that going down that path would detract the reader from the main novel results provided by CATS, i.e. the daily variability of the cloud vertical distribution. To do so we decided not to directly include retrievals based on ISCCP data. Instead, our goal was to verify that CATS retrievals capture the general qualitative feature of the daily cycle of cloud amounts, compared to the baseline dataset usually considered (ISCCP).

C. Stubenrauch, W. B. Rossow, and S. Kinne, 2012: Assessment of global cloud datasets from satellites: A project of the World Climate Research Programme Global Energy and Water Cycle Experiment (GEWEX) Radiation Panel. WCRP Rep. 23/2012, 176 pp.

*17. L539-541: This could also due to the different day-night cloud detection sensitivities between lidar and ISCCP passive measurements.*

This is a possible explanation that we now have included in the article. Thanks.

*18. L574-579: You could try to use CALIOP 1064 only measurements to run the same detection to minimize the difference.*

This could indeed diminish differences between CATS and CALIOP datasets that are related to the instrument's wavelength differences. Many other differences would remain, like laser pulse energy and repetition frequency (20Hz vs. 5kHz), beam width, telescope field of view, sampling rates, performance of optical elements, etc. Moreover, our focus here is on statistics over large regions and seasons, and the different altitude and orbital paths of both missions imply that comparisons will necessarily be statistical in nature — i.e. both datasets document different clouds anyway. Given this, it is unclear what understanding would be gained by going through the exercise suggested here.

The reviewer's suggestion will be useful though for future research aiming to clarify the reasons behind differences between CALIPSO and CATS cloud detections over case studies.

*19. L585 "Cloud Fraction": either use CF or "cloud fraction".*

Thanks for spotting this, we have corrected the error.

*20. Section 5: It will good to have some discussion on the potential limitations here.*

Section 5 now mentions the limitations of cloud fractions retrieved through spaceborne lidar measurements such as CATS and CALIOP, and highlights the problems that still need to be investigated.

**Reviewer 3 comments and replies**

*"The diurnal cycle of cloud profiles over land and ocean between 51S and 51N, seen by the CATS spaceborne lidar from the International Space Station" by Vincent Noel et al.*
*This paper documents the diurnal cycle of the cloud vertical profiles over a large part of the globe, using CATS lidar, operating on the International Space Station. Cloud fractions from different locations, seasons, instruments have been compared, by taking the advantage of this unique dataset. The study is interesting and useful. But it would be better to relate the role of dynamic and thermodynamic processes to the differences of CF found from different conditions, which is not clearly presented. I recommend some modifications to improve the paper before publication.*

We thank the Reviewer for his or her appreciation. Relating the cloud diurnal cycles documented in this paper to the other processes driving the daily evolution of the troposphere (temperature, water vapor) is the focus of a soon-to-be-submitted paper we are currently working on.

**Major issues:**

*1. The second paragraph of Introduction needs more support references to help the readers to better understand the background. For example, 'well documented by passive satellite imagery', it would be better to add in relative works. The same suggestion for 'b) cloud detections from ground-based active instruments' part. There are a lot of works have been done with ground- based instruments on cloud property analysis, I would appreciate if you can give a few references here.*

Our initial idea was that paragraph would sum up the findings of the articles referenced in the previous paragraph ("Those studies…"). However, the first paragraph only references studies of cloud diurnal cycles documented from space, whereas the findings in the second paragraph are general. Following this observation, we have rewritten the first two paragraphs of the introduction, trying to support our assertions with appropriate references.

This comment echoes minor issue #2 from reviewer 2.

*2. The authors chose ARM SGP site for the comparison, is there any particular reason to compare two mid-latitude continental sites with CATS? Comparing with oceanic type of clouds would be interesting, if there are any possibilities. Since you actually didn't really process the SGP dataset, I would suggest that you could keep this part as an additional material.*

Following this comment, and major comment #3 from Reviewer 2, we have tried locating a well-documented, 24/24 dataset of cloud layers covering the period 2015-2017 based on measurements from a ground-based lidar operating in the Tropics, preferably close to the ocean. We have contacted several observatories (e.g OPAR) but it appears lidar-based cloud layer products are often unvalidated and/or suffer from irregular or non-diurnal sampling.

Following the Reviewer's suggestion, we investigated ARM data [1] and found several datasets based on Tropics and/or oceanic measurements and promising cloud layer information. We found that:
- Datasets from Nauru Island and Darwin Australia did not overlap with CATS timeframe
- Datasets from Brazil and Ascension Island cover the CATS timeframe but only contained profiles of Attenuated Backscatter (without cloud detection) — doing the cloud detection ourselves would require external validation
- Only datasets from the ARM Eastern North Atlantic (ENA) atmospheric observatory [2] are close to our criterias above. This site provides cloud layers derived from ground-based lidar measurements made in an oceanic environment, unlike the SIRTA and ARM-SGP datasets considered in the initial article.

Since the ENA observatory is located at 39°N, it is at best sub-tropical. It is however the only oceanic ARM site we found that could provide a 24/24 robust cloud layers dataset covering the CATS time period.

Our initial exploration of the enaarsclkazrbnd1kolliasC1 dataset (based on a combination of lidar and radar data) showed unusual results during the 2017 summer (see figure below). We contacted ARM people, who explained the problem comes from unresolved issues with lidar cloud detections and suggested rebuilding the cloud layers based on the cloud mask source product and ignoring the lidar-only detections. This resolved the problem, but in effect turned it into a radar-based product.

[Figure]

*Figure 1 - Cloud fraction over ARM-ENA for 2017 JJA using both lidar and radar cloud detections (left) and radar-only cloud detections (right)*

The paper now includes (Sect. 3.3) a direct comparison of the diurnal cycles of cloud fraction profiles as documented by CATS over the ENA site and from ground-based radar detections.

We have also obtained the ARM-SGP ground-based lidar+radar cloud detections in order to directly include them in the paper for comparison with CATS data. Exploration of the sgparsclkazrbnd1kolliasC1 dataset showed artefacts similar to those from the ARM-ENA datasets (vertical steps in cloud fraction) when including lidar-only cloud detections. We thus had again to consider only radar detections, leading to results very similar to those presented in Zhao et al. (2016).

The fact that we uncovered issues with those ARM datasets, which are apparently among the most reliable, confirms how difficult it is to obtain faultless lidar-based cloud retrievals over long periods. Providing our own analysis of the data allowed us to only use cloud detections made during the CATS operation period, making time periods consistent across all ground-based comparisons.

The article now includes comparisons between CATS and:
- Lidar-based cloud retrievals from a midlatitude continental site (SIRTA), already described in e.g. Noel et Haeffelin (2006)
- Radar-based cloud retrievals from a midlatitude continental site (ARM-SGP), already described in e.g. Zhao et al. (2016)
- Radar-based cloud retrivals from a subtropical oceanic site (ARM-ENA).

We hope these improvements to our comparisons between CATS and ground-based datasets better highlights the strengths and limitations of spaceborne and ground-based lidar cloud sampling.

[1] https://www.arm.gov/data
[2] https://www.arm.gov/capabilities/observatories/ena

*3. You talked about the pronounced mid-level clouds over continent many times which has been well documented, could you give a more detailed explanation for that, and the role of dynamic and thermodynamic processes.*

The mid-level clouds CATS detects over Africa, South America and Australia, in the North hemisphere in JJA and the South hemisphere in DJF, might very well be Altocumulus clouds that Wang and Sassen (2012) document in the same locations, altitudes and times. This possibility is now discussed in the text.

**Minor issues:**

*1. On Figure 1, is there any way to show the number of samples on the plots as well? In the text, the latitude range is 51S-51N, but on most of the plots, it's 55S-55N. It is better to keep it consistent.*

Following this comment, the map in Figure 1 now shows the number of profiles sampled by CATS over the JJA 2015-2016-2017 period in 2°x2° grid cells. We thank the Reviewer for this suggestion that makes Figure 1 richer in information.

As noted by the Reviewer, the latitude ranges in figures and their legends were incorrect in the initial version of the manuscript we uploaded to ACPD in February. We fixed those in an updated version submitted in March (this version has "over land and ocean" in the title), which became the one available from the ACPD website during the open discussion [1]. We do not know why the reviewers were provided with the non-updated version and regret the confusion.

[1] https://www.atmos-chem-phys-discuss.net/acp-2018-214/acp-2018-214.pdf

*2. The color bar need to be adjusted and extended to greater than 20%, add in unit, and keep the x axis and y axis consistent for the same figure group, specially figure 4.*

Our attempts to increase the maximum to larger cloud fractions led to poor visibility for areas of weak cloud fractions, which are much more frequent and more frequently discussed in the text. Through experimentations, we found that limiting the color bar to a 20% maximum provided the best compromise between keeping variations of weak cloud fractions visible (e.g. at low altitudes) and not masking too many variations in large cloud fractions, for instance in high clouds in tropical summer conditions or low clouds over ARM-

ENA.

All cloud fraction colorbars should now include units (%). We have made sure all axes remain consistent within the same figure groups.

*3. Figure 5, better to label A, B, C... on the subplot for each location.*

We thank the reviewer for this useful suggestion that Figure 5 now implements.

*4. Line 515: Using passive instruments to retrieve the cloud properties is different from active instruments, they don't have the same sensitivity for the thin clouds. It isn't a fair comparison here.*

Our objective here is not to validate or depreciate either one of the detection approaches. All instruments have different sensitivities to different phenomenas. We do not think that confronting retrievals from passive instruments with retrievals from active instruments is unfair to the passive instruments — it shows how each instrument understands a scene, which we think helps the readers familiar with either one, or both, to understand what is actually going on. However, since the sentence in question did not bring any significant value to the manuscript, we have rewritten it to avoid any misunderstanding.

*5. Another thing to note is that, besides cloud detection, retrieving a cloud top height from especially for thin clouds and in multi-layered situations.*

We think some words are missing from the comment. We guess the Reviewer points out that  cloud top heights retrieved from passive measurements can suffer from large uncertainties, especially in presence of thin clouds and multi-layered situations. We agree with his comment.

**Reviewer 4 comments and replies**

*1. The paper discusses the diurnal changes in cloud fraction, but commonly CF is meant to represent the fraction of a grid box area or sensor field of view that is covered in cloud. Would it not be correct to give the results as cloud frequency instead, as that is what is actually being measured?*

During the past years, interactions with co-authors and reviewers led to our adoption of the following naming scheme:
- "cloud cover" to name the fraction of a grid box area covered in cloud
- "cloud fraction profile" to name a vertical profile describing at each altitude level the fraction of shots containing clouds

Many articles use this distinction, for instance Reverdy et al. (2015), Chepfer et al. (2010) referenced in the main article. The following articles use the same naming scheme:
- Chepfer, H., Noel, V., Chiriaco, M., Wielicki, B., Winker, D., Loeb, N. and Wood, R.: The Potential of a Multidecade Spaceborne Lidar Record to Constrain Cloud Feedback, J. Geophys. Res. Atmos., 123(10), 5433–5454, doi:10.1002/2017JD027742, 2018.
- Cesana, G., et al. (2016), Using in situ airborne measurements to evaluate three cloud phase products derived from CALIPSO, J. Geophys. Res. Atmos.,121, 5788–5808, doi:10.1002/2015JD024334.
- Chepfer, H., V. Noel, D. Winker, and M. Chiriaco (2014), Where and when will we observe cloud changes due to climate warming?, Geophys. Res. Lett.,41, 8387–8395, doi:10.1002/2014GL061792
- Reverdy, M., Noel, V., Chepfer, H., and Legras, B.: On the origin of subvisible cirrus clouds in the tropical upper troposphere, Atmos. Chem. Phys., 12, 12081-12101, https://doi.org/10.5194/acp-12-12081-2012, 2012
- Chepfer H., S. Bony, D. Winker, M. Chiriaco, J-L. Dufresne, G. Sèze, 2008: Use of CALIPSO lidar observations to evaluate the cloudiness simulated by a climate model, Geophys. Res. Let., 35, L15704, doi:10.1029/2008GL034207.

We went through the paper to make sure that the paper always mentioned "cloud fraction profile", or specified an altitude range (e.g. "at low altitudes, cloud fractions are high..."). We hope this naming scheme is satisfactory.

*2. Secondly, the figures' color bars max out at CF=20%, with values above 20% visible in many of the figures. It would be good to extend the color bar so that fewer figures saturate like this.*

Our attempts to increase the maximum to larger cloud fractions led to poor visibility for areas of weak cloud fractions, which are much more frequent and more frequently discussed in the text (e.g. at low altitudes). Through experimentations, we found that limiting the color bar to a 20% maximum provided the best compromise between keeping variations of weak cloud fractions visible and not masking too many variations in large cloud fractions, for instance in high clouds in tropical summer conditions or low clouds over ARM-ENA.

[revised manuscript text omitted]

---

## Author Response (AR2)

**The diurnal cycle of cloud profiles over land and ocean between 51°S and 51°N, seen by the**

**CATS spaceborne lidar from the International Space Station**

Vincent Noel, Hélène Chepfer, Marjolaine Chiriaco, John Yorks

Reply to suggestions
June 22nd 2018

*Suggested revisions of figures:*
*- The text in all figures seems rather coarse. Is it possible to improve the resolution of the figures?*
*- Figure 4: please use the same axis labels as in Figure 3. You are currently using Altitude, altitude and height.*
*- Figure 5: Please unify axis labels for this figure as well.*
*- Figure 6: please unify the spacing of the grid lines.*
*- Figure 7: Please remove the figure title*

Following these suggestions, we have updated all figures to make them high-DPI, unified the axis labels and spacing of grid lines, and removed the figure title in Figure 7.